# Cdc7 activates replication checkpoint by phosphorylating the Chk1-binding domain of Claspin in human cells

Chi-Chun Yang[1], Hiroyuki Kato[1], Mayumi Shindo[2], Hisao Masai[1]*

[1]Department of Genome Medicine, Tokyo Metropolitan Institute of Medical Science, Tokyo, Japan; [2]Protein Analyses Laboratory, Tokyo Metropolitan Institute of Medical Science, Tokyo, Japan

**Abstract** Replication checkpoint is essential for maintaining genome integrity in response to various replication stresses as well as during the normal growth. The evolutionarily conserved ATR-Claspin-Chk1 pathway is induced during replication checkpoint activation. Cdc7 kinase, required for initiation of DNA replication at replication origins, has been implicated in checkpoint activation but how it is involved in this pathway has not been known. Here, we show that Cdc7 is required for Claspin-Chk1 interaction in human cancer cells by phosphorylating CKBD (Chk1-binding-domain) of Claspin. The residual Chk1 activation in Cdc7-depleted cells is lost upon further depletion of casein kinase1 (CK1γ1), previously reported to phosphorylate CKBD. Thus, Cdc7, in conjunction with CK1γ1, facilitates the interaction between Claspin and Chk1 through phosphorylating CKBD. We also show that, whereas Cdc7 is predominantly responsible for CKBD phosphorylation in cancer cells, CK1γ1 plays a major role in non-cancer cells, providing rationale for targeting Cdc7 for cancer cell-specific cell killing.

## Introduction

Eukaryotic DNA replication depends on the formation of pre-Replicative Complex on chromatin during the G1 phase of cell cycle, which is mediated by assembly of Orc, Cdc6, Cdt1 and Mcm proteins. Cdc7 kinase plays a crucial role in initiation of DNA replication by phosphorylating Mcm proteins, essential as a part of the replicative helicase (*Masai and Arai, 2002*; *Masai et al., 2010*; *Labib, 2010*).

Inhibition of DNA replication by hydroxyl urea (HU) or UV triggers cellular responses known as replication stress checkpoint (*Branzei and Foiani, 2009*). Conserved checkpoint kinases (Mec1-Rad3-ATR) are activated in response to replication stress, which ultimately activates Rad53-Cds1-Chk1 effector kinases, that inhibits progression of S phase as well as entry into M phase to reduce the genomic instability potentially caused by replication fork arrest. The stalled replication fork caused by replication stress results in activation of the ATR-ATRIP complex and its association with TopBP1, another activator of ATR. Activated ATR phosphorylates Chk1, the crucial step for activation of the key effector kinase. ATR-mediated phosphorylation of Chk1 requires Clapsin as a mediator/adaptor for the signal transfer. Indeed, ATR phosphorylates Chk1 in the Clapsin-Chk1 complex more efficiently than Chk1 alone in the absence of Claspin in virto (*Lindsey-Boltz et al., 2009*).

Cdc7 kinase has been implicated in replication checkpoint responses. In budding yeast, it is known that Rad53 checkpoint effector kinase directly phosphorylates Dbf4 upon fork stalling to block late origin firing most likely through inhibition of the Cdc7 kinase function (*Zegerman and Diffley, 2010*; *Lopez-Mosqueda et al., 2010*; *Duch et al., 2011*), although the mechanism of this regulation is not clear (*Weinreich and Stillman, 1999*). In mammalian cells, it was implicated in repair of stalled replication forks through Rad18 (*Day et al., 2010*). In addition to its role in the

**\*For correspondence:**
masai-hs@igakuken.or.jp

**Competing interests:** The authors declare that no competing interests exist.

**eLife digest** It takes a human cell between six and eight hours to copy all three billion letters of its genome. During this time, any interruption to the process can lead to genetic errors, putting the cell in danger of developing disease. To guard against this, cells use a checkpoint system, testing their own health before, during and after DNA replication to make sure that they are ready for the next step. If a cell detects a problem while copying its DNA, it responds by activating proteins called checkpoint kinases. These stop the cell from continuing until the problem is resolved. One of these checkpoint kinases is a protein called Chk1, which switches on if the cell gets stuck part way through copying its DNA.

To switch Chk1 on, the cell first needs to activate a protein called Claspin. Activating Claspin involves adding a chemical phosphate group to part of the Claspin protein. A third protein takes on this role, but its identity is controversial. Recent research points to a protein called casein kinase 1, but it was also possible that another protein, Cdc7 kinase, might be involved.

To find out, Yang et al. used gene editing to lower the levels of Cdc7 in human cancer cells. The cells were able to copy their DNA under normal conditions, but they struggled to activate Chk1 when DNA replication stopped. Biochemical tests revealed that this was because, without Cdc7, Claspin was not receiving the phosphate group it needed. Even so, the cancer cells still had some Chk1 activation, which meant that they must be able to activate some of their Claspin. So, Yang et al. tried getting rid of both Cdc7 and the other candidate protein, casein kinase 1. This stopped Chk1 activation completely, revealing that although the cancer cells mainly used Cdc7 to activate Claspin, they also used casein kinase 1. In tests on non-cancerous cells, the results were the other way around; healthy cells mainly used casein kinase 1 and relied less heavily on Cdc7.

These differences could prove useful for drug design. One of the challenges in cancer treatment is producing drugs that target cancer cells while leaving healthy cells unharmed. Future research could explore whether blocking Cdc7 could stop Chk1 activation in cancer cells only. This could stop the diseased cells fixing problems with their DNA replication, making it harder for them to survive.

effector phase of checkpoint, the role of Cdc7 kinase in checkpoint activation has been suggested. In fission yeast, activation of checkpoint kinase was impaired in *cdc7* mutant cells (*Shimmoto et al., 2009*; *Matsumoto et al., 2010*). However, a possibility that the reduced number of active replication forks in these mutants is responsible for compromised checkpoint activation could not be ruled out (*Shimada et al., 2002*). However, the impaired checkpoint activation in *cdc7* bypass mutants (Δ*mcm4N* Δ*cdc7* in budding yeast and *rif1Δ hsk1Δ* in fission yeast; *Sheu and Stillman, 2010*; *Hayano et al., 2011*; *Ogi et al., 2008*; our unpublished data) provided strong evidence that Cdc7 is required for checkpoint activation in yeasts. In mammalian cells, induced knockout of Cdc7 gene in mouse ES cells as well as siRNA-mediated inhibition of Cdc7 expression in cancer cells resulted in almost complete loss of Chk1 activation in response to HU or UV irradiation (*Kim et al., 2008*). Later, requirement of Cdc7 for timely checkpoint activation in cancer cells was confirmed by using a compound that inhibits Cdc7 kinase (*Rainey et al., 2013*). Thus, it is now well established that Cdc7 is required for replication checkpoint activation. Furthermore, replication stress-induced hyperphosphorylation of Claspin/Mrc1 exemplified by mobility-shift on PAGE is largely gone in cells where Cdc7 activity is compromised (*Kim et al., 2008*; *Shimmoto et al., 2009*). This suggests a possibility that Cdc7 may regulate checkpoint through Claspin/Mrc1. However, the precise mechanism by which Cdc7 activates replication checkpoint has not been known.

Claspin, originally discovered as a factor that binds to Chk1 in *Xenopus* egg extract (*Kumagai and Dunphy, 2000*), and its yeast homologue, Mrc1, are essential for activation of downstream effector kinases (Chk1 and Cds1/Rad53, respectively), and are required for replication checkpoint control as a mediator (*Chini and Chen, 2003*; *Yoo et al., 2006*; *Lindsey-Boltz et al., 2009*; *Alcasabas et al., 2001*; *Osborn and Elledge, 2003*; *Tanaka and Russell, 2001*). Claspin/Mrc1 is required also for efficient fork progression (*Lin et al., 2004*; *Petermann et al., 2008*; *Scorah and McGowan, 2009*; *Szyjka et al., 2005*). Claspin interacts with various replication factors and other factors including ATR, Chk1, Cdc7 kinase, Cdc45, Tim, MCM4, MCM10, PCNA, DNA polymerases α, δ, ε, And-1, and Rad9 (*Gambus et al., 2006*; *Izawa et al., 2011*; *Lee et al., 2005*; *Brondello et al.,*

*2007*; *Serçin and Kemp, 2011*; *Gold and Dunphy, 2010*; *Uno and Masai, 2011*; *Liu et al., 2012*; *Hao et al., 2015*), as well as with DNA (*Sar et al., 2004*; *Zhao and Russell, 2004*) suggesting its role at the replication forks and potentially in initiation. Yeast Mrc1 was shown to move along with replication fork, linking the helicase components to the replicative polymerases (*Katou et al., 2003*). More recently, Mrc1, in conjunction with Tof1/Csm3, was shown to stimulate DNA replication fork progression in an in vitro reconstitution assay system (*Yeeles et al., 2017*). We recently reported a novel role of Claspin as a recruiter of Cdc7 kinase for efficient phosphorylation of Mcm proteins required for initiation (*Yang et al., 2016*). Cdc7-recruiting function and its potential role in origin firing regulation was reported also for fission yeast Mrc1 (*Matsumoto et al., 2017*; *Masai et al., 2017*).

The role of Claspin/Mrc1 as a replication checkpoint mediator is well established from yeasts to human. In metazoan Claspin, phosphopeptide motifs (CKBD [Chk1-binding domain] or CKAD [Chk1-activating domain]) were identified that are required for regulated binding of Chk1 (*Kumagai and Dunphy, 2003*). In vitro reconstituted system was also reported in which Chk1 activation could be monitored in the presence of ATR (*Lindsey-Boltz et al., 2009*). In *Xenopus* egg extracts, conserved serine-864 and serine-895 are phosphorylated upon replication stress and this phosphorylation is required for checkpoint activation. CKBD is required for checkpoint activation, and a phosphopeptide containing CKBD is sufficient for binding to Chk1. In human cells, Thr-916 residue in CKBD was reported to be phosphorylated in response to HU and the 3A mutant (T916A S945A S983A) is deficient in checkpoint activation (*Chini and Chen, 2006*).

The nature of kinase(s) responsible for CKBD phosphorylation has been controversial. Chk1 kinase was reported to phosphorylate Thr-916 in vitro (*Chini and Chen, 2006*), although it was later reported that Chk1 is not responsible for the phosphorylation of the CKAD in vivo (*Bennett et al., 2008*). On the other hand, Dunphy's group reported that casein kinase is a potential kinase responsible for phosphorylating CKBD (*Meng et al., 2011*). They showed that casein kinases can phosphorylate in vitro the critical threonine residue in CKBD, and that casein kinase $\gamma1$, among different types of casein kinases, promotes phosphorylation of CKBD in vivo and its depletion reduced phosphorylation of Thr-916 of the CKBD polypeptide (899-953) ectopically expressed in human cells, and diminished checkpoint activation.

Here, through CRISPR/Cas9 system, we have established a derivative of HCT116 (human colon cancer cell line) in which the promoter of Cdc7 gene is mutated. This cell line (HCT116-323), expressing Cdc7 at a low level, replicates its DNA at a normal rate, but replication checkpoint activation was significantly reduced. The AP (acidic patch) motif near the C-terminus of Claspin interacts with Cdc7 and is required for Cdc7-mediated phosphorylation of Claspin and also for its interaction with various replication proteins. We have generated a DE/A mutant ($AP_{DE/A}$) of Claspin, in which all the acidic residues in aa988-1086 were replaced by alanine, and have shown that this mutant does not interact with Cdc7 and is not phosphorylated by Cdc7 in spite of the presence of all the serine/threonine residues. We speculated this is due to the inability of the $AP_{DE/A}$ mutant to recruit Cdc7 (*Yang et al., 2016*). We found that $AP_{DE/A}$ is not able to activate Chk1 in response to HU, and cannot bind to Chk1 in spite of the presence of intact CKBD sequence. We surmised this is probably due to the absence of required phosphorylation of CKBD in $AP_{DE/A}$ mutant. Through mass spectrometry analyses, a number of phosphorylation sites were identified near CKBD, including S945 in CKBD. Cdc7 depletion resulted in loss of most of these phosphorylations including S945. These results provide strong evidence for the proposal that Cdc7 plays a crucial role in phosphorylation of CKBD.

We noted that complete suppression of Chk1 activation requires depletion of both Cdc7 and CK1$\gamma$1 in all the cell lines tested. The dependency of Chk1 kinase activation on each kinase differs between cancer cells and non-cancer cells; cancer cells expressing a high level of Cdc7 exhibit higher dependency on Cdc7, whereas non-cancer cells with a lower level of Cdc7 depend more on Chk1$\gamma$1. We propose that either Cdc7 or CK1$\gamma$1 can phosphorylate CKBD for checkpoint activation, while the cellular context affects the pathway choice in different cells.

## Results

### Cdc7 is required for replication checkpoint activation in human cells

Previously, we reported that conditional knockout of Cdc7 in mouse ES cells resulted in loss of Chk1 activation (measured by Chk1 S317 phosphorylation) in response to HU or UV irradiation. We also showed siRNA-mediated inhibition of Cdc7 expression resulted in significant reduction of Chk1 activation in cancer cells (*Kim et al., 2008*). It was later reported that Cdc7 inhibition using an inhibitor delayed the checkpoint activation in cancer cells (*Rainey et al., 2013*). We reexamined this in human cancer cell lines by knocking down Cdc7 expression with siRNA (*Figure 1—figure supplement 1*). In the human colon cancer cell line, HCT116, Chk1 activation was detected at 15 min after addition of 2 mM HU, and increased up to 50 min. At 48 hr after transfection of Cdc7 siRNA, Cdc7 expression was reduced by over 90%. Under this condition, Chk1 activation was reduced by ~90% at 30 min compared to the parental cells.

In order to more precisely examine the effect of the Cdc7 level on the replication checkpoint activation, we generated a derivative of HCT116. By using CRISPR-Cas9, we generated a mutant cell line, HCT116-323, in which Cdc7 expression level was reduced due to a deletion in its promoter region. The deletion covers 12 bp near the transcription initiation site (*Figure 1—figure supplement 2*). The growth of HCT116-323 was slightly slower than the parent (data not shown), but the DNA synthesis or S phase progression was not noticeably affected. There was no significant difference in the level of BrdU incorporation (*Figure 1A*), and S phase proceeded in a similar timing for up to 8 hr after release from the double thymidine block. However, at 10 hr after release, the wild-type HCT116 cells proceeded to G1, while HCT116-323 cells have not divided yet (*Figure 1B*). This suggests that HCT116-323 cells have some difficulty going through mitosis, although eventually they enter mitosis. The Cdc7 protein level was reduced by 90% in HCT116-323 compared to the parent HCT116. After HU treatment, Chk1 activation was reduced by about 80% at 30 min in HCT116-323 compared to the parent (*Figure 1C*), confirming that a certain level of Cdc7 activity is required for full activation of Chk1 in HCT116.

### The acidic patch DE/A mutant of Claspin does not interact with Cdc7 and cannot support Chk1 activation

Claspin is a phosphoprotein, and at least some of these phosphorylations may be mediated by Cdc7 kinase (*Kim et al., 2008*; *Rainey et al., 2013*). We expressed various mutant forms of Claspin tagged with Flag at the C-terminus in 293T (human embryonic kidney cells 293 expressing SV40 large T antigen) cells and pulled down with anti-Flag antibody. We generated ST27A, ST5A and ST19A, in which all the serines/threonines in aa903 ~1120, aa1121 ~1218, and aa1219 ~1337, respectively, were replaced by alanine. Whereas Chk1 was co-pulled down with the wild-type and ST5A and ST19A, it was not pulled down with ST27A (*Figure 2A*, lanes 8, 10–12). This is likely due to the alanine substitution of serine/threonine of CKBD in this mutant, as was shown before (*Kumagai and Dunphy, 2003*; *Meng et al., 2011*; *Chini and Chen, 2006*). We previously reported that the acidic patch (AP; aa988-1086) of Claspin is required for its interaction with Cdc7 kinase. The DE/A mutant of AP ($AP_{DE/A}$) in which all the acidic residues in AP were replaced with alanine is deficient in interaction with Cdc7 and fails to support phosphorylation of Mcm in non-cancer cells (*Yang et al., 2016*). Intriguingly, $AP_{DE/A}$ mutant did not cause defect in phosphorylation of Mcm in cancer cells. In the same pull down assays in 293 T cells, Chk1 was not pulled down with the $AP_{DE/A}$ mutant as well, although CKBD is intact in this mutant (*Figure 2A*, lane 9).

Using the Claspin conditional knockout MEF cells, we examined the ability of the $AP_{DE/A}$ mutant to support Chk1 activation. The wild-type or $AP_{DE/A}$ mutant Flag-tagged Claspin was stably expressed in the mutant MEF cells and endogenous Claspin was knocked out by transfection of Ad-Cre. The Claspin mutant cells without transgene did not show Chk1 activation (*Figure 2B*, lane 2, pChk1), as expected from the loss of Claspin, whereas expression of the wild-type Claspin restored it (*Figure 2B*, lane 4). In contrast, the $AP_{DE/A}$ mutant did not restore the Chk1 activation, consistent with its reduced binding to Chk1 (*Figure 2B*, lane 6). We examined the function of the Claspin $AP_{DE/A}$ mutant for checkpoint activation in U2OS cells (human bone osteosarcoma epithelial cells) as well. The wild-type or $AP_{DE/A}$ mutant Claspin was expressed on a retroviral vector. The endogenous Claspin was knocked down by siRNA specific to its 3'-noncoding segment. The wild-type Claspin could

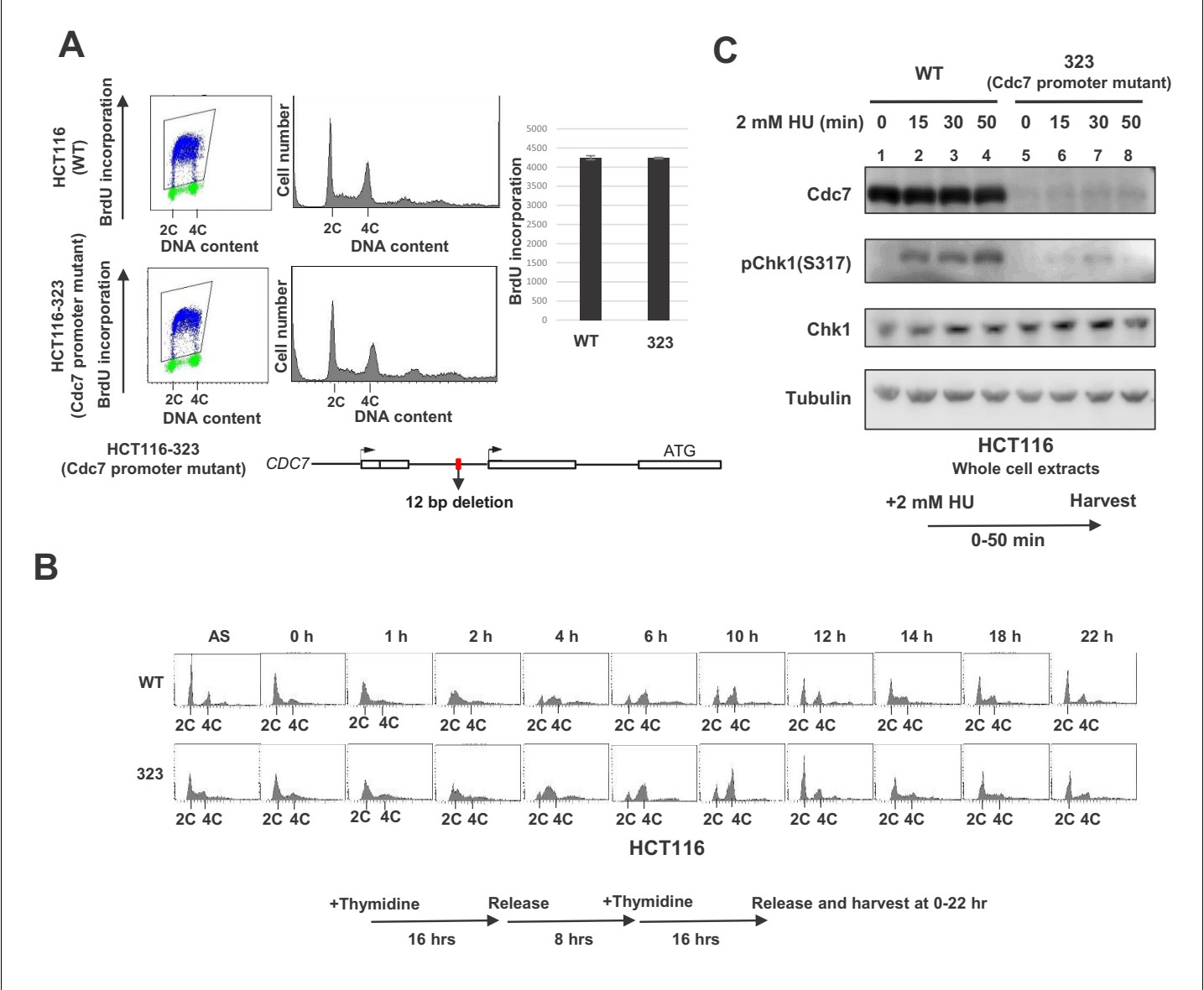

**Figure 1.** Cdc7 is required for activation of Chk1 in response to replication stress. (A) HCT116 and its derivative, HCT116-323, in which the promoter for the Cdc7 gene was mutated (see *Figure 1—figure supplement 1* for details of the mutation), were incubated with BrdU for 20 min and were analyzed by FACS for BrdU incorporation. The average FITC intensity of each cell is shown in the graph. The mean ±s.d. values from three replicates are shown. (B) HCT116 and HCT116-323 cells were arrested at the G1/S boundary by double thymidine block, as outlined in the figure, and were released into the medium without thymidine. Cells were harvested at the times indicated after the release and DNA contents were analyzed by FACS. (C) HCT116 and HCT116-323 were treated with 2 mM HU for 0, 15, 30, 50 min and the whole cell extracts were analyzed by western blotting with indicated antibodies.

The online version of this article includes the following source data and figure supplement(s) for figure 1:

**Source data 1.** Quantification for graph (three independent FACS experiments) in *Figure 1A*.
**Figure supplement 1.** Cdc7 is required for activation of Chk1 in HCT116 cells.
**Figure supplement 2.** Sequences of the promoter region of human Cdc7 gene and deletion introduced by CRISPR-Cas9.

restore the Chk1 phosphorylation in Claspin-depleted cells, whereas the $AP_{DE/A}$ mutant did not fully restore it (*Figure 2C*, lanes 10 and 12). We previously showed that $AP_{DE/A}$ is not phosphorylated by Cdc7 in vivo as well as in vitro, presumably due to impaired recruitment of Cdc7 to Claspin (*Yang et al., 2016*). Thus, the above results are consistent with the notion that Cdc7-mediated phosphorylation of Claspin is required for checkpoint activation.

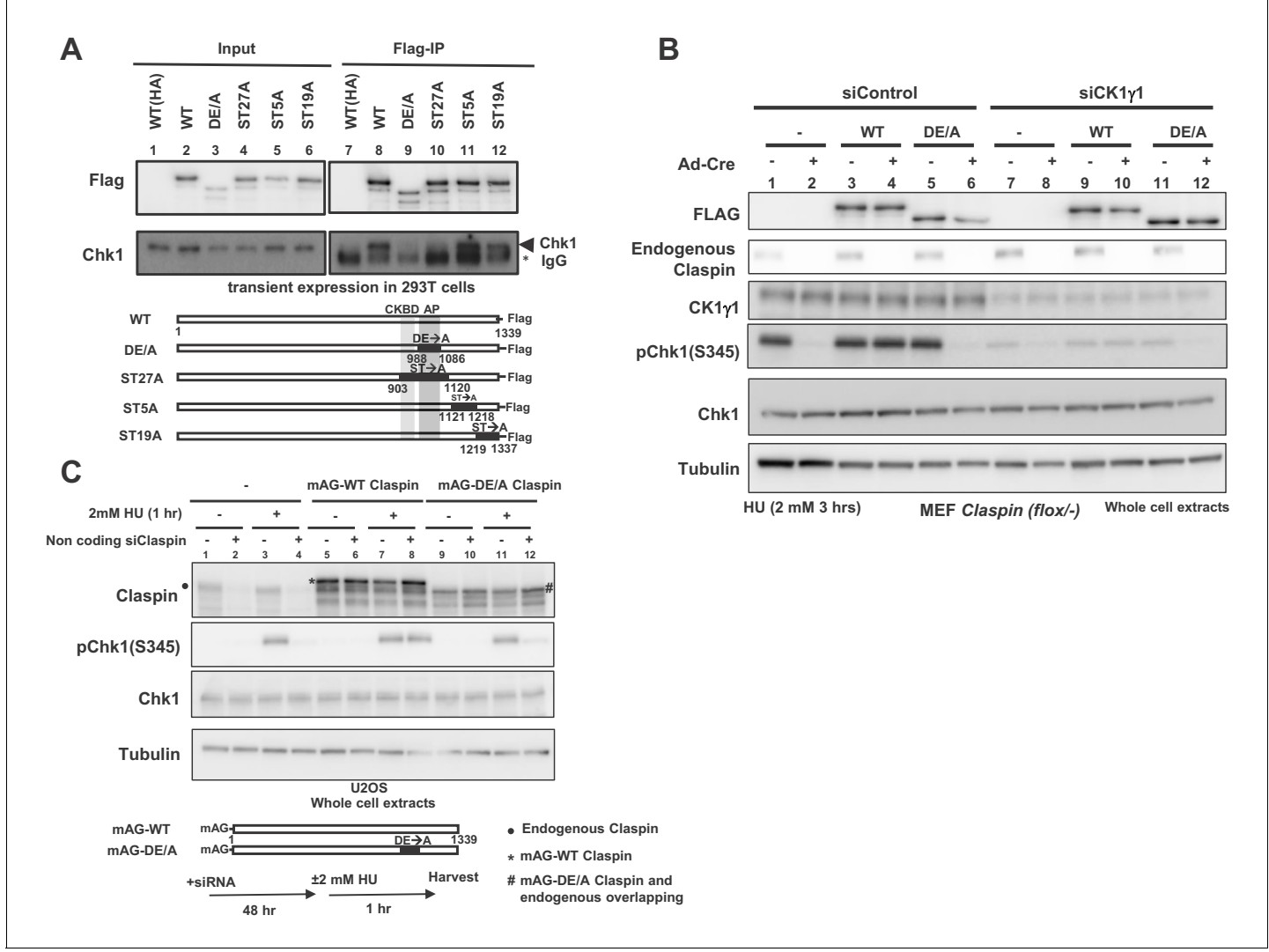

**Figure 2.** AP (Acidic Patch) of Claspin is required for interaction with Chk1 and Chk1 activation. (A) C-terminally Flag-tagged wild type (WT) and mutant *Claspin* indicated were transiently expressed in 293 T cells and were pulled down by anti-Flag beads. Immunoprecipitated proteins were analyzed by western blotting with anti-Flag and anti-Chk1 antibodies. Shown below the panels are schematic diagrams of the mutant Claspin proteins. Black boxes indicated the segments in which amino acid replacements were introduced. Asterisks indicate IgG; arrowheads indicate Chk1. WT(HA) represents the HA-tagged Claspin, which was used as a negative control. (B) Stable clones of *Claspin* flox /-MEF cells expressing the Flag-tagged wild-type or AP$_{DE/A}$ mutant Claspin or no transgene were infected with Ad-Cre for 48 hr or non-treated. Cells were then treated with CK1γ1 siRNA (lanes 7–12) or non-treated (lanes 1–6) for 24 hr and then with 2 mM HU for 3 hr, and the whole cell extracts were analyzed by western blotting with indicated antibodies. (C) U2OS cells were infected by retroviruses expressing mAG-WT Claspin or mAG-AP$_{DE/A}$ Claspin (momeric Azami Green fluorescent protein fused at the N-terminus of Claspin, which does not affect the functionso of Clapsin). At 2 days after infection, non-coding Claspin siRNA or control siRNA was introduced into the cells for two days, and then the cells were treated with 2 mM HU for 1 hr or non-treated before harvest. The whole cell extracts were analyzed by western blotting with indicated antibodies. mAG indicates monomeric Azami Green fluorescent protein.

The online version of this article includes the following figure supplement(s) for figure 2:

**Figure supplement 1.** Pull down assays with biotinylated CKBD phospho-oligopeptides.

## Claspin interacts with Chk1 in a Cdc7-mediated phosphorylation-dependent manner

CKBD was discovered as a motif to which Chk1 specifically binds (*Kumagai and Dunphy, 2000*). This binding was reported to be dependent on the phosphorylation of the conserved threonine or serine present in the CKBD. We have confirmed phospho-dependent interaction by pull down assays. Biotinylated CKBD-containing phosphopeptides (NC [aa901-955 with phospho-Thr-916 and

phospho-Ser-945], N [aa905-925 with phospho-Thr-916] and C [aa934-954 with phospho-Ser-945]) were mixed with the extract from 293 T cells and the pulled-down materials were examined by western analyses. Chk1 was efficiently recovered with NC but not with N or C (*Figure 2—figure supplement 1A*; compare lanes 4, 6 and 8). When the biotinylated NC oligopeptide was pretreated with λ-phosphatase, binding was completely lost (*Figure 2—figure supplement 1A*, lane 5). Similar results were obtained with the purified Chk1 protein (*Figure 2—figure supplement 1B and C*). This is consistent with previous findings, and indicates that at least two phosphorylated CKBD boxes are required for efficient Chk1 binding (*Kumagai and Dunphy, 2003*; *Chini and Chen, 2006*). If the dephosphorylated oligopeptide was incubated with Cdc7 prior to pull down, the level of Chk1 pull down increased (*Figure 2—figure supplement 1C*, compare lanes 8 and 9). The recovery is not complete, because the efficiency of Cdc7-mediated phosphorylation is inefficient on the NC oligopeptide due to the absence of AP, the recruiter of Cdc7 kinase.

This suggests a possibility that Cdc7 phosphorylates CKBD and promotes the interaction of Claspin with Chk1. We therefore examined the effect of Cdc7 on the interaction of Claspin with Chk1 in cells. 293 T cells were either treated with Cdc7 siRNA or non-treated and stimulated by 2 mM HU for 4 hr. Cell extracts were prepared and immunoprecipitation was conducted by using anti-Chk1 antibody. Claspin was coimmunoprecipitated with Chk1, and the amount of the precipitated Claspin increased after HU (*Figure 3A*, lanes 5 and 6). In Cdc7-depleted cells, the amount of the precipitated Claspin significantly decreased (*Figure 3A*, lanes 7 and 8), indicating that the Claspin-Chk1 interaction depends on Cdc7 in 293 T cells.

We have conducted similar experiments by using Cdc7 promoter mutant cell line (HCT116-323). Immunoprecipitation with anti-Chk1 antibody pulled down the endogenous Claspin protein in the wild-type HCT116 treated with HU (*Figure 3B*, lane 3). However, the amount of coimmunoprecipitated Claspin was significantly reduced in HCT116-323 in which the Cdc7 protein level was reduced (*Figure 3B*, lane 4). These results establish that binding of Chk1 to Claspin requires Cdc7 function in HCT116 as well.

## Cdc7 is responsible for phosphorylation of CKBD in 293 T cells

Above in vitro and in vivo (cellular) results strongly suggest that Cdc7 phosphorylates CKBD for recruitment of Chk1. To evaluate this possibility in more detail, we conducted mass spectrometry analyses of Claspin after treatment with HU in 293 T cells. 293 T cells, either transfected with Cdc7 siRNA or control siRNA, were treated with 2 mM HU for 24 hr or non-treated. Cell lysates prepared with RIPA buffer were subjected to immunoprecipitation with anti-Claspin antibody. The precipitated Claspin bands were detected on SDS-PAGE (*Figure 4—figure supplement 1*) and were subjected to mass spectrometry analyses.

In the control siRNA transfected cells with HU, we detected 31 phosphorylated amino acids, 25 of which are present in the C-terminal half of the protein. In the siCdc7 transfected cells with HU, 35 phosphorylated amino acids were detected, 21 of which are present in the C-terminal half of the protein. Furthermore, 20 or 14 phosphorylated amino acids were detected on the 266-amino acid segment (720 ~ 985) overlapping with a part of the CKBD segment in the control siRNA or siCdc7 transfected cells with HU, respectively. 11 phosphorylated serines and threonines in the 266 amino acid segment disappeared after transfection of siCdc7 in comparison with the control siRNA (with HU). Thr-916 and Ser-945 were among those that disappeared after Cdc7 depletion (*Figure 4* and *Figure 4—figure supplement 1B*). These results support the conclusion that Cdc7 plays a major role in phosphorylating CKBD for checkpoint activation in 293 T cells.

## Casein kinase also contributes to the phosphorylation of CKBD

Dunphy's group previously reported that casein kinase 1 γ1 (CK1γ1) can promote interaction between Claspin and Chk1 through phosphorylating CKBD in cancer cells (*Meng et al., 2011*). We evaluated the potential role of CK1γ1 in replication checkpoint in conjunction with Cdc7.

When CK1γ1 was reduced in U2OS by siRNA, Chk1 activation was reduced by about half, consistent with the previous report (*Figure 5A*, lanes 9–12). Cdc7 knockdown reduced the Chk1 activation by about 80%, but still weak Chk1 S317 signal remained (*Figure 5A*, lanes 5–8). In Cdc7-CK1γ1 double knockdown, Chk1 activation was almost completely gone (*Figure 5A*, lanes 13–16). Codepletion

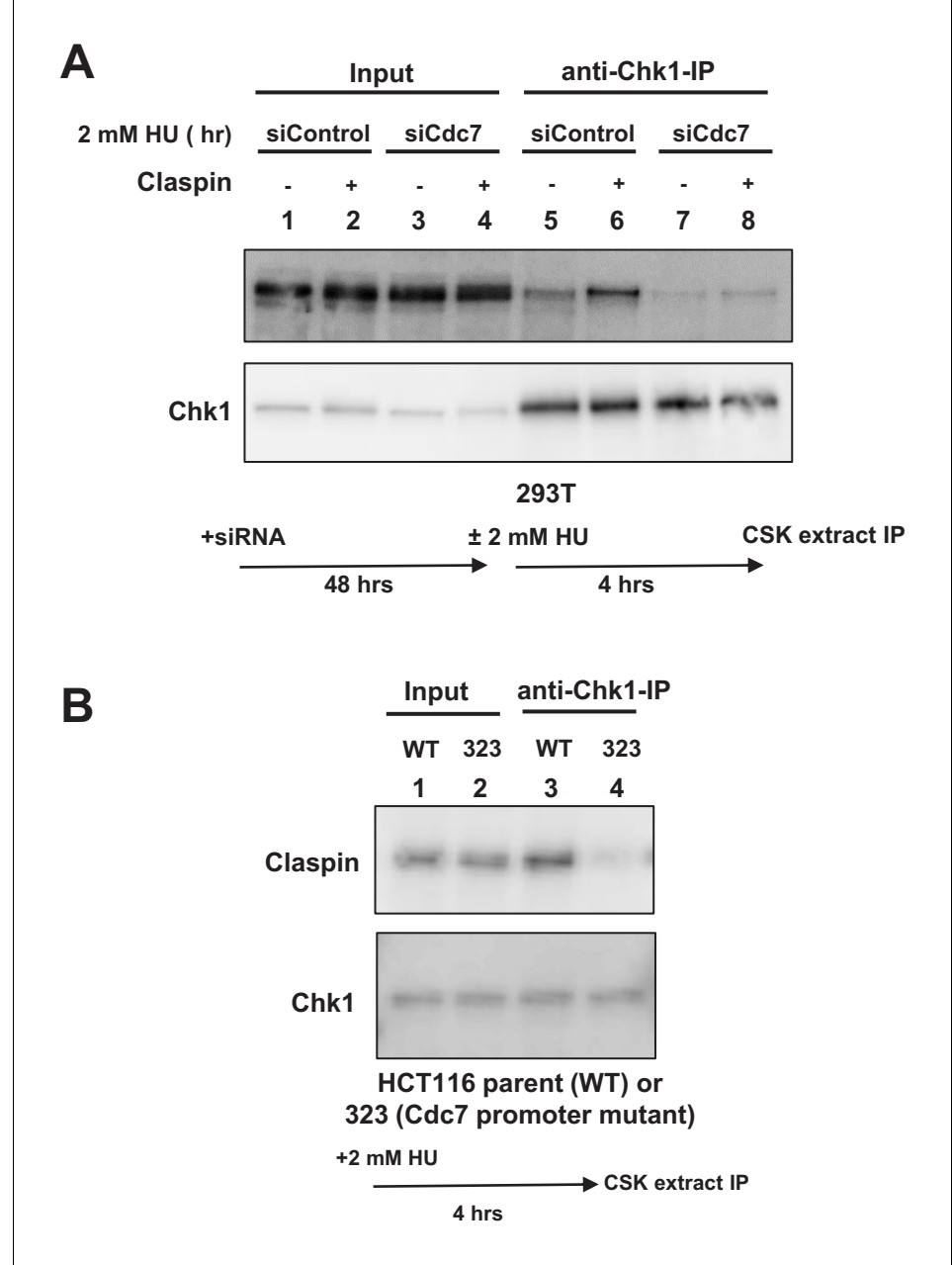

**Figure 3.** Cdc7 is required for interaction of Claspin with Chk1. (**A**) 293 T cells transfected with siControl or siCdc7 for 48 hr were treated with 2 mM HU for 4 hr or non-treated. Cells were lysed with CSK buffer containing 0.1% Triton-X100 and proteins were pulled down with anti-Chk1 antibody and were analyzed by western blotting with anti-Claspin and anti-Chk1 antibodies. (**B**) HCT116 and its derivative, HCT116-323 (Cdc7 promoter mutant), were treated with 2 mM HU for 4 hr and proteins were analyzed as in **A**.

of Cdc7 and CK1γ1 led to additive effect on decrease of Chk1 activation in HeLa (human cervix epitheloid carcinoma) cells as well (*Figure 5—figure supplement 1*).

CK1γ1 depletion resulted only in partial suppression of Chk1 activation also in HCT116 (*Figure 5B*, lanes 5–8). However, that in HCT116-323, where Chk1 activation is reduced due to decreased Cdc7 expression, almost completely suppressed it, suggesting the residual Chk1 activation in HCT116-323 is mediated by CK1γ1 (*Figure 5B*, lanes 13–16). BrdU incorporation of HCT116-323 was not significantly affected by CK1γ1 depletion (*Figure 5—figure supplement 2*), suggesting that suppression of Chk1 activation in HCT116-323 by siCK1γ1 is not caused by reduced DNA

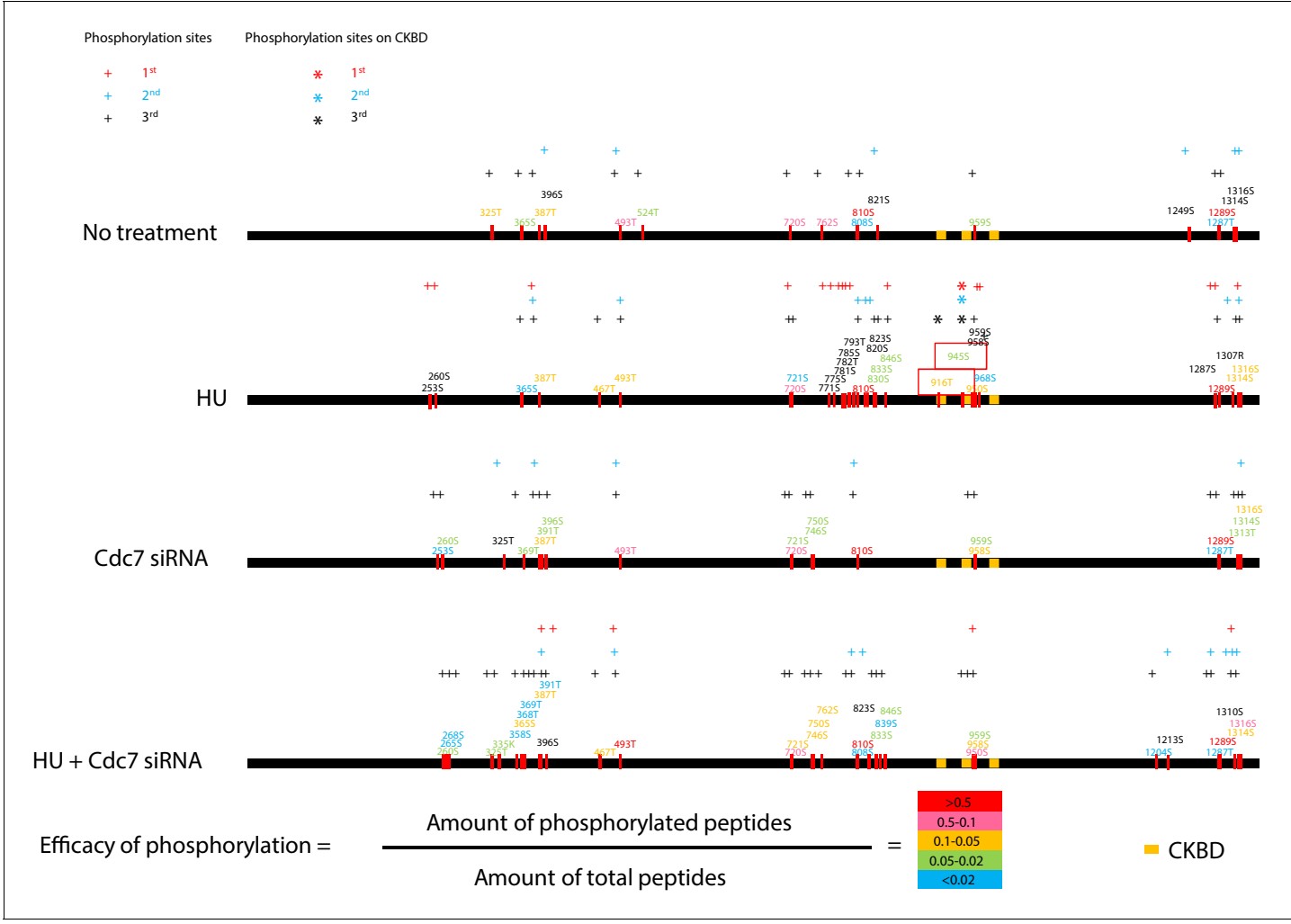

**Figure 4.** Cdc7 is required for phosphorylation of CKBD of Claspin in 293 T cells. Phosphorylation sites of Claspin in the non-treated, HU-treated, siCdc7-treated and siCdc7+HU-treated cells were determined by mass spectrometry analyses and are shown on the drawing along the polypeptide. See *Figure 4—figure supplement 1* for experimental details. 'HU' and 'HU+Cdc7 siRNA' experiments were repeated three times, and 'No treatment' and 'Cdc7 siRNA' experiments were repeated two times (2nd and 3rd). Phosphorylation sites identified are indicated by the small bars and amino acid numbers, which are color-coded by the efficacy of phosphorylation, as determined by the formula in the figure. The efficacy of phosphorylation is based on the third experiment. The amino acid numbers in black are the phosphorylated residues detected only in the 1st and/or 2nd experiments. Three CKBD are shown by yellow boxes. Putative phosphorylation sites in the CKBD are boxed in red. '+" indicate phosphorylation sites detected in each experiment. Those on CKBD detected are indicated by '*". '+' and '*' are color coded in red, blue and black which stands for 1st, 2nd and 3rd experiment, respectively.

The online version of this article includes the following source data and figure supplement(s) for figure 4:

**Source data 1.** Primary data (1st experiment) for *Figure 4*.
**Source data 2.** Primary data (2nd experiment) for *Figure 4*.
**Source data 3.** Primary data (3rd experiment) for *Figure 4*.
**Figure supplement 1.** Mass spectrometry analyses of phosphorylation sites on Clapsin in 293 T cells.

replication. All these results strongly suggest that both Cdc7 and CK1γ1 play roles in Chk1 activation in response to replication stress and complete suppression of Chk1 activation requires inactivation of both kinases.

In cancer cells, we consistently observed that Cdc7 depletion reduced Chk1 activation more than CK1γ1 depletion did. Indeed, cell death measured by TUNEL assays increased after siRNA transfection targeted either to Cdc7 or CK1γ1. HU treatment increased TUNEL positive cells in Cdc7 siRNA treated cells but not in CK1γ1 siRNA treated cells (*Figure 5—figure supplement 3A and B*, compare row 2 and 10 or 3 and 11 in **B**). Unexpectedly, in NHDF (normal human dermal cells), CK1γ1

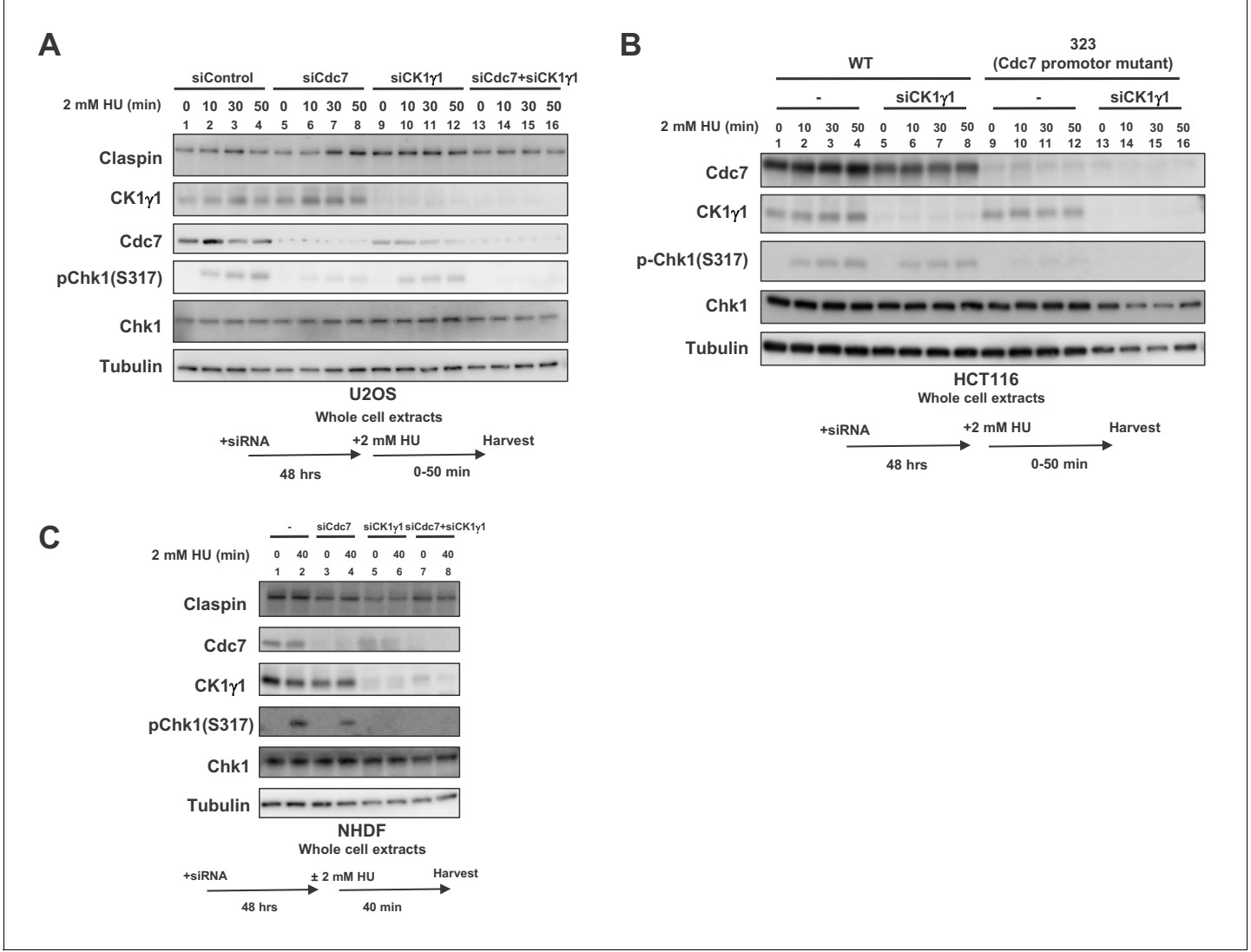

**Figure 5.** Both Cdc7 and CK1γ1 are required for full activation of Chk1 in response to replication stress. (A) U2OS cells were transfected with siControl, siCdc7, siCK1γ1 or siCdc7+siCK1γ1 for 48 hr. (B) HCT116 and HCT116-323, a Cdc7 promoter mutant cell line, were transfected with siControl or siCK1γ1 for 48 hr. In A and B, cells were harvested at 0, 10, 30 and 50 min after addition of 2 mM HU. (C) NHDF cells were transfected with siControl, siCdc7, siCK1γ1 or siCdc7+siCK1γ1 for 48 hr. Cells were non-treated or treated with 2 mM HU for 40 min. In A-C, the whole cell extracts were analyzed by western blotting with indicated antibodies.

The online version of this article includes the following source data and figure supplement(s) for figure 5:

**Figure supplement 1.** Cdc7 and CK1γ1 are required for full activation of Chk1 in response to replication stress in HeLa cells.

**Figure supplement 2.** Effect of CK1γ1 depletion on DNA replication in HCT116-323 cells.

**Figure supplement 2—source data 1.** Quantification for graph (three independent FACS experiments) in *Figure 5—figure supplement 2*.

**Figure supplement 3.** Cdc7 plays a major role in prevention of replication stress-induced cell death in U2OS cells.

**Figure supplement 3—source data 1.** Quantification for graph (three independent FACS experiments) in *Figure 5—figure supplement 3*.

**Figure supplement 4.** CK1γ1 plays a major role for full activation of Chk1 in response to replication stress in non-cancer cells.

**Figure supplement 5.** Cdc7 plays a major role for full activation of Chk1 in response to replication stress in 293 T cells 293 T cells were transfected with siControl, siCdc7, siCK1γ1 or siCdc7+siCK1γ1 for 48 hr.

depletion almost completely impaired Chk1 activation (by 90%), whereas Cdc7 depletion only partially (by 50%) inhibited it (*Figure 5C*, lanes 6 and 4). In MEF cells, Chk1 activation was inhibited by more than 80% by CK1γ1 depletion (*Figure 2B*, compare lanes 1 and 7). Similarly, in other non-cancer cells RPE-1 (human epithelial cells immortalized with hTERT) and TIG-3 (a normal diploid fibroblast cell line), depletion of CK1γ1 significantly reduced Chk1 activation (by 90% for both), whereas

that of Cdc7 reduced it by 60% in both cells (*Figure 5—figure supplement 4*, lanes 6 and 7), supporting our conclusion that CK1γ1 plays a major role in checkpoint activation in non-cancer cells.

## Phosphorylation of CKBD by Cdc7 and CK1γ1 in vitro

How do Cdc7 and CK1γ1 collaborate for checkpoint activation? One possibility is that CK1γ1 acts as a priming kinase for subsequent phosphorylation by Cdc7. Phosphorylation of a substrate by Cdc7 kinase has been shown to be facilitated by the 'priming phosphorylation' by other kinases in a number of instances (*Masai et al., 2000*; *Sasanuma et al., 2008*; *Wan et al., 2008*; *Francis et al., 2009*; *Murakami and Keeney, 2014*). These possibilities were examined by conducting the in vitro phosphorylation assays using CKBD containing polypeptides.

We first conducted phosphorylation assays using a Claspin-derived polypeptide (aa897-1100) as a substrate. This substrate contains both AP and CKBD. Both Cdc7 and CK1γ1 phosphorylated this polypeptide in a dose-dependent manner. Addition of Cdc7 in the presence of a low amount of CK1γ1 increased the phosphorylation level of the polypeptide. At a low level of Cdc7, the effect of CK1γ1 was additive on the level of the phosphorylation, whereas, at the highest concentration of Cdc7, similar levels of phosphorylation were observed regardless the presence or absence of CK1γ1 (*Figure 6—figure supplement 1*). CK1γ1 alone also could achieve the similar level of phosphorylation when added in a sufficient amount (data not shown). These results are consistent with the notion that Cdc7 and CK1γ1 can phosphorylate this polypeptide in an independent fashion.

In order to more directly examine whether Cdc7 phosphorylates CKBD, we have prepared the 50 amino acid polypeptide (906 ~ 955) containing two CKBD (wild-type) and their derivatives containing amino acid substitutions at selected serine/threonine residues, and used them as substrates for in vitro phosphorylation assays with Cdc7 and/or CK1γ1. Cdc7 was able to phosphorylate the 50 aa polypeptide (*Figure 6*, lanes 2, 13–15), and this phosphorylation was significantly reduced by alanine substitutions of threonine and serine in CKBD (CKBD-A; *Figure 6*, compare lanes 15 and 16), indicating that CKBD is the major target of Cdc7 kinase on this polypeptide. To assess the roles of other serine/threonine residues on the polypeptide, we replaced all other serine/threonine residues with either alanine (Others-A) or glutamic acid (Others-E). Cdc7 was able to phosphorylate Others-A, albeit at a reduced level compared to Wild-type (*Figure 6*, lanes 15 and 19), whereas Others-E was phosphorylated by Cdc7 at about three times more efficiently than was Wild-type (*Figure 6*, lanes 15 and 22). This is probably due to the effect of acidic residues that facilitate the substrate recognition by Cdc7. In contrast, CK1γ1 did not very efficiently phosphorylate Wild-type (*Figure 6*, lanes 6, 10–12). Consistently, addition of CK1γ1 did not stimulate Cdc7-mediated phosphorylation of Wild-type was (*Figure 6*, lanes 2–5). CK1γ1 alone did not phosphorylate Others-E very efficiently, either (*Figure 6*, lanes 23). These results are consistent with the conclusions that Cdc7 phosphorylates preferentially the serine/threonine residues in CKBD in vitro and that CK1γ1 does not serve as a priming kinase for phosphorylation of CKBD by Cdc7. They also suggest that Cdc7 is more active than CK1γ1 as a CKBD-phosphorylating kinase in vitro.

## CKBD is phosphorylated by Cdc7 or CK1γ1 in a cellular context-dependent manner

These results indicate that Cdc7 and CK1γ1 independently phosphorylate CKBD, most notably at the Thr-916/Ser-945. Then, how is the kinase selected for phosphorylation of CKBD in cells? We examined other cell lines for dependency of checkpoint activation on Cdc7 (*Figure 7—figure supplement 1*). We used XL413, a Cdc7-specific inhibitor, to inhibit Cdc7 kinase. In HL60 (human promyelocytic leukemia cell line), KM12-Luc (human colon cancer cell line), SK-BR-3-Luc (human breast cancer cell line) and HCT15 (human colorectal adenocarcinoma cell line), pretreatment with XL413 reduced HU-induced Chk1 S317 phosphorylation by 32, 25, 24% and 61%, respectively. In contrast, in NCI-H1975-Luc (human lung adenocarcinoma cell line) and NUGC-3 (human gastric cancer cell line), the effect of XL413 was weaker (15% and 4% inhibition, respectively). We noted that the levels of Cdc7 were significantly lower in these latter cell lines than in other cancer cell lines (*Cheng et al., 2013*). Non-cancer cells including NHDF, RPE-1, and TIG-3 exhibited more dependency on CK1γ1 than on Cdc7 for checkpoint activation (*Figure 5C* and *Figure 5—figure supplement 4*).

We therefore examined the levels of Cdc7 and CK1γ1 proteins in different cell lines. By comparing the western signals of the whole cell extracts from a fixed number of various cells with those of

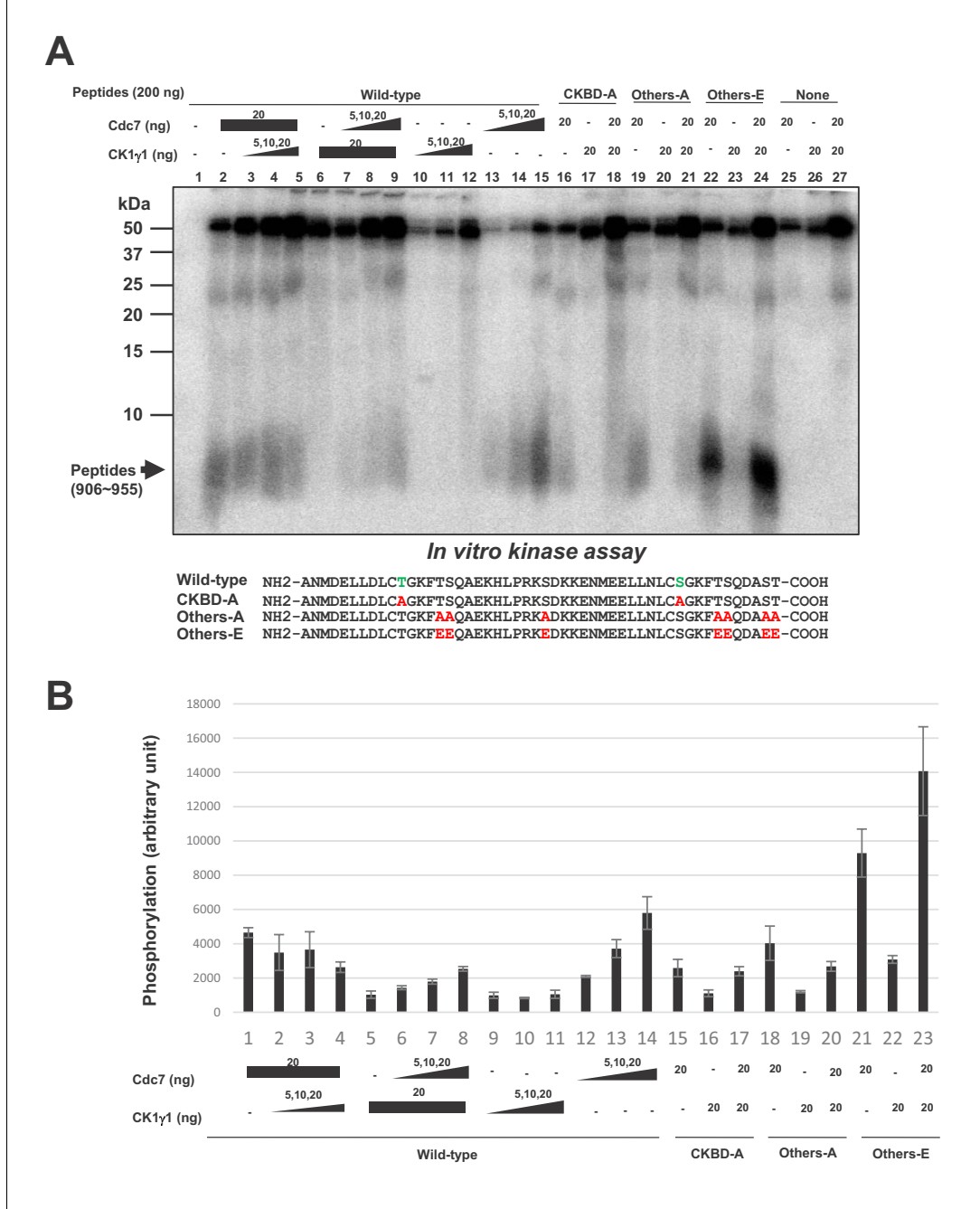

**Figure 6.** Cdc7 phosphorylates T916 and S945. (**A**) Two hundred ng (34 pmole) each of wild type CKBD peptide (wild-type) or mutant peptides (CKBD-A, Others-A and Others-E) was incubated in the kinase reaction with indicated amounts of purified Cdc7-ASK and/or CK1γ1 in the presence of [γ-$^{32}$P]-ATP for 60 min at 30°C. Samples were analyzed on 18% SDS-PAGE. After silver staining, the gel was autoradiographed. The position of the substrate peptides in the gel is indicated by an arrow. The amino acid sequences of each peptide are shown below the panel. The serine and threonine in CKBD are in green, and the mutated residues are shown in red. (**B**) Quantification of the phosphorylation level of the peptide in (**A**). The averages of three independent kinase assays are shown with error bars.

The online version of this article includes the following source data and figure supplement(s) for figure 6:

**Source data 1.** Quantification for graph (three independent FACS experiments) in *Figure 6B*.

**Figure supplement 1.** Cdc7 and CK1γ1 phosphorylate Claspin independently.

**Figure supplement 2.** CK1γ1 phosphorylates Claspin in a manner independent of serine/threonine residues in the CKBD-AP segment.

the control protein samples of known quantity, we estimated the numbers of Cdc7 and CK1γ1 molecules per cell. The results indicate that the numbers of Cdc7 molecules in cancer cells are 10–50 fold more than those in non-cancer cells. On the other hand, the numbers of the CK1γ1 molecules in non-cancer and cancer cells are roughly the same ($1.4 \sim 2.4 \times 10^5$; except for HeLa cells which have four-fold more than NHDF; *Figure 7—figure supplement 2*). The results suggest a possibility that Cdc7 predominantly serves as a CKBD kinase in the majority of cancer cells, since it is overexpressed, whereas in NHDF, CK1γ1 is more abundant than Cdc7, and thus serves as a major kinase for CKBD phosphorylation.

In order to test the above possibility, we examined the effects of manipulation of protein expression levels on the checkpoint pathway choice. In HCT116, CK1γ1 depletion does not significantly affect the Chk1 activation, whereas Cdc7 depletion reduced it by over 50% (*Figure 7A*, lanes 2 and 3). In contrast, in HCT-323 that expressed Cdc7 at a reduced level, CK1γ1 depletion reduced the Chk1 activation almost completely (*Figure 7A and B*, compare lanes 5 and 7). We also examined whether overexpression of Cdc7 modulates the effect of CK1γ1 depletion. Cdc7 was expressed in NHDF by a lentivirus expression vector and the cells were adjusted for 3 days. Then, CK1γ1 or Cdc7 was depleted by siRNA for two days. HU-induced Chk1 S317 phosphorylation was reduced by more than 70% by CK1γ1 depletion in mock-transfected cells (*Figure 7C and D*, lanes 1 and 3), but the inhibition was mitigated in Cdc7-expressing cells (*Figure 7C and D*, lanes 5 and 7), whereas stronger inhibition by Cdc7 siRNA (by ~60%) was observed in Cdc7-overexpressing NHDF cells (*Figure 7C and D*, lanes 5 and 6). This effect appears to depend on the 'adaptation' period. Incubation of Cdc7-overexpressing cells for three weeks before addition of siRNA resulted in even better dependence on Cdc7 for checkpoint activation (*Figure 7—figure supplement 3*). These results are consistent with the idea that the relative levels of Cdc7 and CK1γ1 are one of the factors that determine the pathway for replication stress-induced Claspin phosphorylation.

## Discussion

Cdc7 kinase plays a conserved and crucial role in initiation of DNA replication by converting the pre-Replicative Complex into an active replicative complex through phosphorylation of Mcm, the essential helicase component. Potential roles of Cdc7 in checkpoint activation were reported in yeast (*Takeda et al., 2001*), but effects of Cdc7 downregulation on origin firing have made it difficult to draw a definitive conclusion on the role of Cdc7 in checkpoint activation. Generation of bypass mutations of *cdc7/hsk1* in yeasts permitted a precise test on the role of Cdc7 in replication checkpoint separate from its function in initiation. These studies showed that Cdc7 is required for activation of checkpoint (*Sheu and Stillman, 2010*; Shimmoto and Masai, unpublished data). Similarly, in mammalian cells, Cdc7 is very likely to be required for efficient and timely Chk1 activation, as shown by using Cdc7 knockout cells, siRNA-mediated Cdc7 knockdown or a low molecular weight Cdc7 inhibitor (*Kim et al., 2008*; *Rainey et al., 2013*). However, precise mechanisms of Cdc7-mediated checkpoint activation have remained unclear.

Claspin and its yeast homologue, Mrc1, play a crucial role in replication checkpoint as a mediator (*Kumagai and Dunphy, 2000*; *Alcasabas et al., 2001*; *Tanaka and Russell, 2001*). Claspin recruits Chk1 through CKBD. CKBD needs to be phosphorylated to interact with Chk1. Chk1 and CK1γ1 have been reported to phosphorylate CKBD (*Chini and Chen, 2006*; *Meng et al., 2011*), although the in vivo role of Chk1 in phosphorylation of Claspin has been ruled out (*Bennett et al., 2008*). In contrast, CK1γ1 is required for efficient checkpoint activation, and evidence reported by Dunphy's group indicates a potential role of CK1γ1 in phosphorylation of CKBD. In this report, we have presented evidence that Cdc7, a kinase crucial for initiation of DNA replication, plays a role in phosphorylation of CKBD, thus contributing to checkpoint activation. We also show that Cdc7 and CK1γ1 phosphorylate CKBD independently and additively. Intriguingly, we found that two kinases contribute to CKBD phosphorylation differentially in cancer and non-cancer cells in a manner dictated by the levels of both kinases.

### Cdc7 phosphorylates CKBD of Claspin for checkpoint activation

The following results presented in this study show that Cdc7 is predominantly responsible for phosphorylation of CKBD of Claspin in response to replication stress in most of the cancer cells.

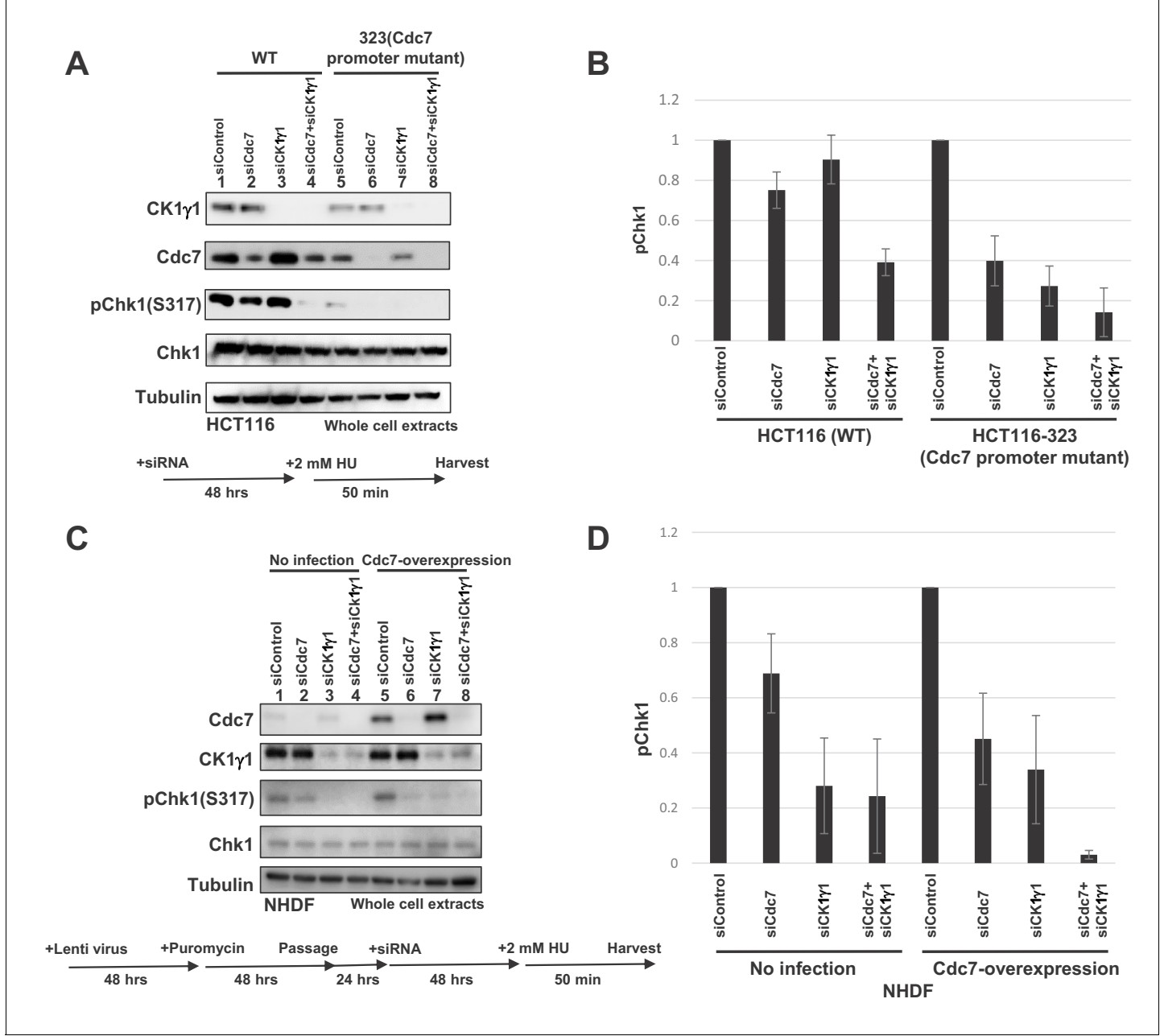

**Figure 7.** Levels of Cdc7 protein affect the dependency on CK1γ1 for checkpoint activation. (A) HCT116 and HCT116-323 (Cdc7 promoter mutant) cells were transfected with siControl, siCdc7, siCK1γ1 or siCdc7+ siCK1γ1 for 48 hr. Cells were then treated with 2 mM HU for 50 min. Whole cell extracts were analyzed by western blotting with indicated antibodies. (B). Quantification of the level of Chk1 phosphorylation in (A). (C) NHDF cells were infected by Cdc7-expressing lentiviruses. At 2 days after infection, cells were selected by puromycin (4 μg/ml) for 2 days, and were transfected with indicated siRNA for 2 days, followed by incubation with 2 mM HU for 50 min. Whole cell extracts were analyzed by western blotting with indicated antibodies. (D) Quantification of the level of Chk1 phosphorylation in (C). In B and D, the level of Chk1 phosphorylation in the siControl sample was taken as 1, and relative levels of the phosphorylation in other siRNA samples are shown for each cell line. Averages of the three independent experiments are shown with error bars.

The online version of this article includes the following source data and figure supplement(s) for figure 7:

**Source data 1.** Quantification for western data (three independent experiments of *Figure 7A*) in *Figure 7B*.
**Source data 2.** Quantification for western data (three independent experiments of *Figure 7C*) in *Figure 7D*.
**Figure supplement 1.** Effect of Cdc7 inhibition on checkpoint activation in various cancer and normal cell lines (related to *Figures 1* and *4* as well).
**Figure supplement 2.** Estimation of the numbers of molecules of Cdc7 and CK1γ1 in various cell lines.
**Figure supplement 3.** Overexpression of Cdc7 in NHDF cells increases dependency on Cdc7 for checkpoint activation.

1. Cdc7 depletion by siRNA or by hypomorphic mutation results in decreased interaction between Chk1 and Claspin in response to replication stress.
2. $AP_{DE/A}$ mutant of Claspin, which does not interact with Cdc7 and is not phosphorylated by Cdc7, cannot activate checkpoint.
3. CKBD of Claspin is phosphorylated by Cdc7 in vitro.

Mass spectrometry analyses revealed the presence of the clusters of phosphorylation sites in and near CKBD, and notably at Thr-916/Ser-945, conserved serines/threonines in the two CKBD. The phosphorylation of Thr-916/Ser-945 is not detected in Cdc7-depleted cells consistently in three independent experiments (*Figure 4* and *Figure 4—figure supplement 1*). These two CKBD appear to be important for binding to Chk1, since the peptide containing both CKBD bound to Chk1 but not the peptide containing only one (*Figure 2—figure supplement 1B*). Using human in vitro cell-free system, mutation of Thr-916 and Ser-945 was shown to inhibit its association with Chk1 (*Clarke and Clarke, 2005*). Our analyses indicate that Cdc7 recruited to AP (aa988 ~ 1086) phosphorylates multiple serine/threonine residues in the adjacent segment (aa720 ~968). These sites do not match very well with previously reported phosphorylation sites of Claspin phosphorylated by Cdc7 in vitro (*Rainey et al., 2013*; the phosphorylation sites do not include serine/threonine in CKBD; only S721 and S744 in aa720 ~ 968). Claspin may adopt a conformation that is more favorable for recruitment of Cdc7 kinase in vivo, that may permit more localized phosphorylation in aa720 ~ 968. Cdc7 likes acidic environment, and thus initial phosphorylation may trigger further recruitment of Cdc7, causing autoactivating loop of phosphorylation. Indeed, the CKBD-derived peptide, in which serine-threonine residues other than Thr-916 and Ser-945 were replaced by glutamic acid, was phosphorylated by Cdc7 with much better efficiency in vitro (*Figure 6*). On the other hand, some Cdc7-dependent phosphorylation sites outside CKBD were detected only in one or two experiments out of the three independent experiments (*Figure 4*). This may suggest promiscuous or stochastic nature of phosphorylation of Claspin by Cdc7 recruited to AP.

## Contribution of CK1γ1 to phosphorylation of CKBD and checkpoint activation

CK1γ1 were previously reported to phosphorylate CKBD (*Meng et al., 2011*). Casein kinase is similar to Cdc7 in its acidophilic preference for substrate selection. Thus, it is predictable that casein kinase also recognizes CKBD which is embedded in acidic environment. In fact, CK1γ1 can phosphorylate CKBD+AP containing polypeptides with efficiency similar to that of Cdc7 (*Figure 6—figure supplement 1*). We examined the possibility that CK1γ1 may initially act as a priming kinase, which phosphorylates serine/threonine residues near the Cdc7 target site. This priming phosphorylation would create acidic environment which may facilitate the recognition by Cdc7. However, we were unable to detect priming phosphorylation events on the aa897-1100 polypeptide containing CKBD used in this study. Both kinases appear to independently and additively phosphorylate this polypeptide in vitro (*Figure 6* and *Figure 6—figure supplement 1*). This is consistent with the fact that, in cancer cells, Cdc7 depletion inhibits checkpoint activation more vigorously than CK1γ1 depletion does, and that combination of both depletions results in almost complete inhibition of checkpoint activation.

Although Cdc7 can phosphorylate Thr-916/Ser-945 on the 50aa oligopeptide (906 ~ 955) in vitro, albeit to a lower extent, CK1γ1 was not able to efficiently phosphorylate it (*Figure 6*). CK1γ1 may need the adjacent polypeptide segment to efficiently recognize Claspin. Furthermore, Cdc7 only poorly phosphorylates ST27A Claspin, in which all the serines/threonines in aa903 ~1120 containing CKBD were replaced by alanine (*Yang et al., 2016*). CK1γ1 efficiently phosphorylates the same polypeptide, suggesting that CK1γ1 can phosphorylate Claspin outside of aa903-1120 (*Figure 6—figure supplement 2*).

## Potential contribution of other factors to Claspin-Chk1 interaction

The full-length Claspin protein binds to purified Chk1 protein in vitro. This in vitro binding occurs with both $AP_{DE/A}$ and or ST27A (data not shown), whereas binding of the 55 aa CKBD polypeptide to Chk1 completely depends on phosphorylation of Thr-916/Ser-945. On the other hand, either $AP_{DE/A}$ or ST27A do not bind to Chk1 in vivo by immunoprecipitation (*Figure 2A*), suggesting better specificity for the phosphorylated CKBD in the cells.

And-1 was reported to promote the interaction between Claspin and Chk1, thereby stimulating efficient Chk1 activation by ATR (*Hao et al., 2015*). ATR-ATRIP was also shown to be required for Claspin-mediated Chk1 activation in *Xenopus* egg extracts (*Kumagai et al., 2004*). Thus, other factors are likely to improve the specificity and efficacy of the Claspin-Chk1 interaction in cells.

## Cell type-dependent differential contribution of Cdc7 and Casein kinase to checkpoint activation

In cancer cells examined, Cdc7 kinase plays a major role in mediating checkpoint activation (*Figure 5A and B*, *Figure 5—figure supplement 1* and *Figure 5—figure supplement 5*). In contrast, in non-cancer cultured cells, such as NHDF, PRE-1 or TIG-3, CK1γ1 plays a predominant role for checkpoint activation (*Figure 5C* and *Figure 5—figure supplement 4*). The Claspin $AP_{DE/A}$ mutant cannot support checkpoint activation even in MEF cells (*Figure 2*), which depends more on CK1γ1 for checkpoint activation (*Figure 2B*). This is due to the inability of the $AP_{DE/A}$ mutant to interact with CK1γ1 as well (data not shown).

The selection of a kinase for checkpoint activation appears to be correlated with the relative levels of the two kinases in given cells, since reduction of the Cdc7 level in HCT116 increased dependency on CK1γ1 (*Figure 7*), while overexpression of Cdc7 in normal cells decreased dependency on CK1γ1 (*Figure 7* and *Figure 7—figure supplement 3*).

Cdc7 inhibition or depletion reduces DNA synthesis both in cancer and non-cancer cells. However, it induces cell death in the former cells but not in the latter (*Montagnoli et al., 2006*; *Montagnoli et al., 2008*; *Sawa and Masai, 2008*; *Ito et al., 2008*; *Ito et al., 2012*). Our results indicate higher dependency on Cdc7 for checkpoint activation in cancer cells than in non-cancer cells. The checkpoint defect by the loss of Cdc7 specifically observed in cancer cells may explain cancer cell-specific cell death induction by Cdc7 inhibition (*Ito et al., 2008*; *Ito et al., 2012*), since failure to properly respond to replication stress would be directly linked to genome instability.

In summary, our findings in this report show that Cdc7 is a major kinase that mediates phosphorylation of CKBD of Claspin in response to replication stress, especially in Cdc7-overexpressing cancer cells. This finding also clarifies the role of Cdc7 in replication checkpoint activation. Another important finding of this report is that complete shutdown of Chk1 activation requires loss of both Cdc7 and CK1γ1, and that contribution of each kinase on Chk1 activation may vary from one cell-type to another and may be at least partly determined by the abundance of each protein and cellular context. The crucial role of Cdc7 in checkpoint activation in cancer cells but not in non-cancer cells may provide molecular explanation for cancer cell-specific cell death induced by Cdc7 inhibition.

## Materials and methods

### Cell culture

293T, HeLa, NHDF, HCT116 and U2OS were obtained from ATCC. *Claspin flox /- Mouse Embryonic Fibroblasts (MEFs)* were established from E12.5 embryos (*Yang et al., 2016*). *Claspin flox /- MEFs* stably expressing the wild-type or $AP_{DE/A}$ mutant Claspin were established by infecting recombinant retroviruses expressing these cDNAs (*Yang et al., 2016*). Cells were grown in Dulbecco's modified Eagle's medium (high glucose) supplemented with 15% fetal bovine serum (NICHIREI), 2 mM L-glutamine, 1% sodium pyruvate, 100 U/ml penicillin and 100 μg/ml streptomycin in a humidified atmosphere of 5% $CO_2$, 95% air at 37°C. HCT-15-Luc#1, NCI-H1975-Luc, NUGC-3 were grown in RPMI supplemented with 10% fetal bovine serum (Gibco); HL-60 were grown in RPMI supplemented with 55 μM β- 2-mercaptoethanol and 10% fetal bovine serum (Gibco); 5K-BR-3-Luc were grown in McCoy's 5A supplemented with 10% fetal bovine serum (Gibco); KM12-Luc, RPE-1 and TIG-3 were grown in Dulbecco's modified Eagle's medium supplemented with 10% fetal bovine serum (Gibco).

### Antibodies and proteins

Anti-Chk1 (sc-8408 or sc-7898, 1:200) and anti-MCM2 (sc-9839, 1:200) antibodies were from Santa Cruz Biotechnology. Anti-MCM2 S53 (A300-756A, 1:1000) antibody was from Bethyl. Anti-Cdc7 (K0070-3, 1:1000), anti-Chk1 (K0085-3, 1:1000) and anti-Flag (M185-3L, 1:1000) antibodies were from MBL. Anti-CK1γ1 was from Biorbyt (orb37898,1:2000). Anti-Chk1 S317 (#2341, 1:1000) and anti-tubulin (T5168, 1:1000) antibodies were from Cell Signaling and Sigma-Aldrich, respectively;

anti-human Claspin, anti-mouse Claspin anti-MCM4, and anti-MCM4 S6T7 antibodies were previously described (*Masai et al., 2006*; *Yang et al., 2016*). Purified Cdc7-ASK (05–109), Chk1 (02–117) and CK1γ1(03–105) kinase were from Carna Bioscience, Inc Cdc7-ASK was also purified from Sf9 cells as previously described (*Masai et al., 2000*).

## Plasmid construction for expression of wild-type and AP$_{DE/A}$ mutant of Claspin

The Claspin-coding segment (*XhoI/XbaI* fragment) of CSII-EF MCS-mAG (monomeric Azami Green; *Karasawa et al., 2003*)-TEV-6His-Claspin-3Flag, CSII-EF MCS-6His-Claspin-3Flag or CSII-EF MCS-6His-Claspin-HA plasmid DNA was replaced by DNA fragments encoding portions of Claspin or its mutant forms, obtained by PCR amplification, to express truncated or mutant forms of Claspin. The *Eco*RI-*Hpa*I fragment containing the wild-type or AP$_{DE/A}$ mutant Claspin DNA from mAG-TEV-6His-Claspin-3xFlag was inserted at the *Eco*RI/*Sna*BI site of pMX-IP (Addgene) to construct retroviral expression vectors.

## Construction of Cdc7 promoter mutant cells derived from HCT116

A short deletion was introduced into the promoter segment of the Cdc7 gene basically according to the method previously described (*Shalem et al., 2014*). Briefly, target sequence (CR1, CR2, CR3, or CR4; see *Figure 1—figure supplement 2* for sequences) was inserted into the lentiCRISPR v2 vector, and the lentivirus carrying the target sequence was prepared. A colorectal cancer cell line HCT116, which is pseudodiploid and has two copies of the Cdc7 gene (https://cansar.icr.ac.uk/cansar/cell-lines/HCT-116/copy_number_variation/chromosome_1/), was used for targeting. Forty-eight hours after infection, 4 µg/ml of puromycin was added for 4 days and resistant cells were plated in 96 well plates to isolate single cell-derived clones. Cdc7 expression in each clone was examined by western blot analysis and genomic DNA sequences were determined with PCR products of the targeted segment and also with the plasmids after cloning of the PCR fragments.

## Expression of recombinant proteins in 293 T cells

1.6 µg of expression plasmid DNA in 100 µL of 150 mM NaCl were mixed with 100 µl of 150 mM NaCl supplemented with 7 µl 1 mg /mL PEI (polyethylenimine 'MAX' [MW25,000; Cat.24765; Polyscience, Inc.]). After 30 min incubation at a room temperature, the solution was added to 293 T cells cultured in six well plates with 2 ml of fresh D-MEM in each well (*Uno et al., 2012*).

## Knockdown of Claspin, Cdc7 or CK1γ1 by siRNA

HeLa, U2OS and HCT116 cells were transfected siRNA by oligofectamine (Thermo Fisher Scientific) for 48 hr. NHDF cells were transfected siRNA by lipofectamine 2000 (Thermo Fisher Scientific) for 48 hr. The siRNA sequences are provided in *Supplementary file 1* or *Supplementary file 2*.

## Western blotting

Proteins in SDS-sample buffer were incubated at 96°C for 1 min, were run on 4 ~ 20% gradient SDS-PAGE (ATTO) and then transferred to Hybond ECL membranes (GE Healthcare), followed by western blot analysis with the indicated antibodies. Blots were then incubated for 1 hr with the secondary antibody conjugated to horseradish peroxidase. Detection was conducted with Lumi-Light PLUS Western Blotting Substrate (Roche) and images were obtained with LAS3000 (Fujifilm).

## Growth curves

$1 \times 10^5$ cells of HCT116 and its Cdc7 promoter mutant derivative were plated in 6-well plates. Cells were incubated and cell numbers were counted at day 1, 2, 3, 4, and five by staining with Trypan blue.

## BrdU incorporation

BrdU was added to HCT116 and Cdc7 promoter mutant cells in 6-well plates at 20 µM for 20 min. Cells were harvested and were fixed at −20°C by 75% ethanol. After wash with the wash buffer (0.5% bovine serum albumin in phosphate-buffered saline), cells were treated with 2 N HCl for 20 min and then with 0.1 M sodium borate (pH8.5) for 2 min at room temperature. Cells were then

treated with FITC-conjugated anti-BrdU antibody (BD biosciences, 51–23,614 l) for 20 min at room temperature in the dark, and further incubated with propidium iodide (25 µg/ ml) and RNaseA (100 µg/ ml) for 30 min at room temperature, followed by analyses with FACS (fluorescence-activated cell sorter; *Yoshizawa-Sugata and Masai, 2007*).

## Pull-down with biotinylated CKBD phospho-oligopeptides

Two pmole of biotinylated phosphopeptides, treated with λPPase or not, were mixed with cell lysates or with purified Chk1 protein, pulled down by streptavidin beads. After washing, samples were analyzed by western blotting using anti-Chk1 antibody. Purified Cdc7 protein and ATP were also added together with purified Chk1 protein when the template peptides needed to be phosphorylated.

## Cell fractionation and immunoprecipitation

293 T cells transiently expressing wild-type or mutant Claspin proteins (C-terminally tagged with 3x-Flag) were lysed in CSK buffer (10 mM PIPES-KOH [pH6.8], 100 mM potassium glutamate, 300 mM sucrose, 1 mM MgCl$_2$, 1 mM EGTA, 1 mM DTT, 1 mM Na$_3$VO$_4$, 50 mM NaF, 0.1 mM ATP, protease inhibitor-PI tablet [Roche] and 0.5 mM PMSF), containing 0.1% TritonX-100 and 10 units/ ml Benzonase (Amersham plc.). After rotating for 60 min in cold room, the supernatants were incubated with anti-Flag M2 affinity beads (SIGMA, A2220) for 60 min at 4˚C. The beads were washed with CSK buffer three times and proteins bound to the beads were analyzed by western blotting.

For immunoprecipitation of Claspin, 293T or HCT116 cells were transfected with indicated siRNA or control siRNA for 48 hr, and were further treated with 2 mM HU for indicated time or non-treated. Cells were lysed by CSK buffer as above and supernatants were mixed with anti-Claspin antibody conjugated to Dynabead protein G (Invitrogen) at 4˚C for 1 hr. After wash with PBS with 0.01% Tween-20, the proteins on beads were separated on 4–20% gradient gel and analyzed by western blotting with anti-Claspin and anti-Chk1 antibodies.

## Protein purification

293 T cells (two 10 cm dishes) transfected with a plasmid expressing wild-type or mutant/truncated versions of Claspin for 40 hr were lysed with CSK buffer as above. The proteins of the supernatants were mixed anti-Flag M2 affinity beads (Sigma-Aldrich) and washed by Flag wash buffer (50 mM NaPi [pH7.5], 10% glycerol, 300 mM NaCl, 0.2 mM PMSF and PI tablet). The bound proteins were eluted by Flag elution buffer (50 mM NaPi [pH7.5], 10% glycerol, 30 mM NaCl, 200 µg /ml 3xFlag peptide [SIGMA], 0.1 mM PMSF and PI tablet) (*Uno et al., 2012*).

## Pull down assays with purified proteins in vitro

To examine the interaction between the purified Claspin (C-terminally Flag-tagged) and Chk1, the recombinant wild-type or mutant Claspin proteins were incubated with purified Chk1 (Carna Bioscience) in a reaction buffer (40 mM HEPES-KOH [pH 7.6], 20 mM K-glutamate, 2.5 mM MgCl$_2$, 1 mM EGTA, 0.01% TritonX-100, 1 mM Na$_3$VO$_4$, 50 mM NaF, 1 mM ATP, 0.5 mM PMSF and PI tablet) at 4˚C for 1 hr. Anti-Flag M2 affinity beads were added and beads were recovered by centrifugation, washed twice with the reaction buffer. Proteins attached to the beads were analyzed by SDS-PAGE and detected by western blotting.

To examine the effect of Cdc7 on interaction between Claspin and Chk1, purified Claspin protein was first incubated for 30 min at 30˚C with λPPase to remove any preexisting phosphates, and Claspin was recovered with anti-Flag M2 affinity beads. After thoroughly washing the beads with washing buffer (25 mM HEPES-KOH [pH 7.6], 0.01% TritonX-100, 0.5 mM PMSF and PI tablet), the recovered Claspin attached to anti-Flag M2 affinity beads was mixed with Chk1 alone or Chk1 and Cdc7 in the same reaction buffer at 30˚C for 30 min and at 4˚C for 1 hr. After washing three times with the washing buffer, proteins attached to the beads were analyzed by SDS-PAGE and detected by western blotting.

## Analyses of proteins in *Claspin flox* /- MEFs stably expressing wild type and AP$_{DE/A}$ mutant Claspin

$1 \times 10^5$ of *Claspin flox* /- MEFs or *Claspin flox* /- MEFs stably expressing the wild type or AP$_{DE/A}$ mutant Claspin were infected with Ad-Cre for 48 hr or not treated. Cells, treated with 2 mM HU for 3 hr, were harvested, were resuspended in sample buffer and the whole cell extracts were analyzed by SDS-PAGE, followed by western blotting analyses.

## Mass spectrometry analyses of Claspin

293 T cells (15 cm dish, five plates) were transfected with Cdc7 siRNA or control siRNA for 24 hr, and 2 mM HU was added or non-treated for 24 hr. Cells were lysed by RIPA buffer (50 mM Tris-HCl [pH8.0], 150 mM NaCl, 0.5 % w/v sodium deoxycholate, 1.0 % NP-40 and 0.1% SDS) containing 1 mM DTT, 1 mM Na$_3$VO$_4$, 50 mM NaF, 0.1 mM ATP, 0.5 mM PMSF and PI tablet. The supernatants were mixed with anti-Claspin antibody conjugated to Dynabead protein G (Invitrogen) at 4°C for 1 hr. After washing with PBS containing 0.01% Tween-20, the proteins on beads were separated on 4–20% gradient gel and stained by CBB. Claspin protein was extracted from the gel, digested by Trypsin and phospho-threonines or -serines were analyzed by mass spectroscopy. The results are presented in Supplementary spreadsheet.

## In vitro kinase assays

Claspin-derived polypeptides (100 ng) or CKBD peptides (200 ng) were incubated with the Cdc7-ASK complex and/or CK1γ1 in the kinase reaction buffer (40 mM Hepes-KOH [pH7.6]; 2.5 mM spermine; 5 mM MgCl2; 0.5 mM EGTA; 0.5 mM EDTA; 1 mM Na$_3$VO4; 1 mM NaF; 2 mM DTT; 10 μM ATP; 1 μCi [γ-ATP]) for 1 hr at 30°C. One-fourth volume of 5x sample buffer was added, heated at 75°C for 1 min and analyzed by SDS-PAGE, followed by silver staining and autoradiogram.

## TUNEL assay

For cell death analyses, DNA fragmentation was analyzed by TUNEL assay using In Situ Direct DNA Fragmentation Assay Kit (Abcam) according to the manufacturer's protocol. In brief, $1 \times 10^5$ U2OS cells were transfected with siRNA for 72 hr, and treated with 2 mM HU for 0, 24 or 48 hr. Harvested cells were fixed by 1% paraformaldehyde and washed by PBS, followed by addition of 70% ice-cold ethanol for 30 min. After washing by wash buffer, cells were mixed with staining solution for 1 hr at 37°C and rinsed by the rinse buffer. After removing the rinse buffer by centrifugation, cells were resuspended in Propidium Iodide/RNase A solution and incubated in the dark for 30 min at room temperature. The FITC labeled cells were analyzed by FACS.

## Estimation of numbers of Cdc7 and CK1γ1 molecules in various cell lines

Whole cell extracts prepared from $5 \times 10^4$ cells of 293T, HeLa, HCT116, U2OS and NHDF were run on SDS-PAGE along with known amounts of purified Cdc7 and CK1γ1 proteins and were analyzed by western blotting using the antibodies against these proteins.

## Expression of proteins using lentivirus and retrovirus vectors

293 T cells were transfected with the lentivirus vector expressing Cdc7 (on CSII-CMV-puro), pMDLg/pRRE, pRSV-Rev and pMD2.G. Platinum-A (Plat-A) cells were transfected by the retrovirus vector expressing mAG-WT Claspin or mAG- AP$_{DE/A}$ Claspin (on pMX-IP). The fusion of a fluorescent protein at the N-terminus of Claspin does not affect its functions. In both cases, virus-containing medium were harvested at 2 days (for lentivirus) or 3 days (for retrovirus) after transfection and the viruses were concentrated by centrifugation. The concentrated Cdc7-expressing lentiviruses were used for infection of NHDF. The Claspin-expressing retroviruses were for infection of U2OS cells. In each case, at 2 days after infection, puromycin (4 μg/ml) was added and resistant cells were selected for 2 days.

## Cell lines

Cells used were authenticated cell lines obtained from ATCC or JCRB Cell Bank. Mycoplasma contamination tests gave negative results on all the cells used.

## Statistical analyses

Error bars represent the mean ± s.d. values calculated from three independent replicate experiments.

## Acknowledgements

This work was supported by JSPS KAKENHI (Grant-in-Aid for Scientific Research (A) [Grant Numbers 23247031 and 26251004] and Grant-in-Aid for Scientific Research on Priority Areas ['non-coding RNA' and 'Genome Adaptation'; Grant Numbers 24114520 and 25125724, respectively] to HM and by the Naito Foundation Continuation Subsidy for Outstanding Projects (to HM). We also would like to thank the Takeda Science Foundation for the financial support of this study. We would like to thank all the members of the laboratory for helpful discussion. We would like to thank Jun Horiuchi for critical reading of the manuscript.

## Additional information

### Funding

| Funder | Grant reference number | Author |
| --- | --- | --- |
| Japan Society for the Promotion of Science | 23247031 | Hisao Masai |
| Japan Society for the Promotion of Science | 26251004 | Hisao Masai |
| Japan Society for the Promotion of Science | 24114520 | Hisao Masai |
| Japan Society for the Promotion of Science | 25125724 | Hisao Masai |
| Naito Foundation | Continuation Subsidy for Outstanding Projects | Hisao Masai |
| Takeda Science Foundation | | Hisao Masai |

The funders had no role in study design, data collection and interpretation, or the decision to submit the work for publication.

### Author contributions

Chi-Chun Yang, Data curation, Formal analysis, Validation, Investigation, Methodology, Writing - original draft; Hiroyuki Kato, Mayumi Shindo, Investigation, Methodology; Hisao Masai, Conceptualization, Resources, Data curation, Formal analysis, Supervision, Funding acquisition, Validation, Investigation, Writing - original draft, Project administration, Writing - review and editing

### Author ORCIDs

Hisao Masai (ID) https://orcid.org/0000-0003-1268-5302

### Decision letter and Author response

Decision letter https://doi.org/10.7554/eLife.50796.sa1
Author response https://doi.org/10.7554/eLife.50796.sa2

## Additional files

### Supplementary files

• Supplementary file 1. Key Resources Table.

• Supplementary file 2. List of the siRNA sequences used in this study.

• Transparent reporting form

## Data availability

All data generated or analyzed during this study are included in the manuscript and supporting files. Figure 1-source data 1 has been provided for Figure 1A. Figure 4 -source data 1-3 have been provided for Figure 4. Figure 5-figure supplement 2-source data 1 has been provided for Figure 5-figure supplement 2. Figure 5-figure supplement 3-source data 1 has been provided for Figure 5-figure supplement 3B. Figure 6-source data 1 has been provided for Figure 6B. Figure 7-source data 1 has been provided for Figure 7B. Figure 7-source data 2 has been provided for Figure 7D.

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
