## [Decision Letter]

**Acceptance summary:**

The response of cells to DNA damage and DNA replication stress is a major area of investigation, and signal transduction related to these processes differs between cancer and normal cells. Multiple protein kinases have been reported to be involved in the activation of the Chk1 kinase in human cells, as part of the ATR-Claspin mediated signally pathway. In this report, the authors have demonstrated that Cdc7 kinase, which is known to play a key role in the initiation of DNA replication, binds to Claspin-Chk1 and is critical for Chk1 activation. Interestingly, they find that Cdc7 kinase is more important in cancer cells, whereas the previously identified CK1γ1 kinase is important in non-cancer cells. As such the paper clarifies the literature on Chk1 signalling mechanisms.

**Decision letter after peer review:**

Thank you for submitting your article "Cdc7 activates replication checkpoint by phosphorylating the Chk1 binding domain of Claspin" for consideration by *eLife*. Your article has been reviewed by three peer reviewers, including Bruce Stillman as the Reviewing Editor and Reviewer #1, and the evaluation has been overseen by Anna Akhmanova as the Senior Editor.

The reviewers have discussed the reviews with one another and the Reviewing Editor has drafted this decision to help you prepare a revised submission.

Summary:

The Masai lab has previously reported that Claspin helps recruit the Cdc7 kinase to chromosomes to promote the phosphorylation of MCM and the initiation of DNA replication. Specifically, in Yang et al., 2016, they conclude that Cdc7 phosphorylates amino acids within the Chk1 binding domain (CKBD) of Claspin to prevent an intramolecular Claspin interaction that inhibits Claspin's ability to interact with DNA and PCNA.

In this study, they expand upon this published work by providing evidence that the Cdc7-Claspin interaction, again involving phosphorylation of the Chk1 binding domain (CKBD) of Claspin, is also important for activating the ATR-replication stress pathway (via permitting a Claspin-Chk1 interaction). While several reports provide evidence that the Cdc7 kinase functions in activating the ATR-replication stress (ATR-RS) checkpoint, the precise mechanistic explanation has been unclear. Furthermore, there was some confusion on this issue because the CK1γ1 kinase could also activate the ATR-RS checkpoint. This study makes an important contribution by providing evidence that Cdc7 and CK1γ1 independently and additively phosphorylate the Chk1 binding domain (CKBD1) of Claspin to activate the replication stress checkpoint, but that each kinase provides different levels of contributions depending on cell type and levels of the kinase expressed. Cdc7 kinase, due at least in part to its higher expression levels in cancer cell lines, is more critical for ATR-RS checkpoint activation in these cell lines compared to CK1γ1. This latter conclusion has important potential translational implications for considering Cdc7 as a target for cancer therapies.

The evidence that Cdc7 directly interacts with the AP of Claspin and directly phosphorylates the Claspin CKBD is the weakest link in the study as it is difficult to glean how data were quantified. Even in the Yang et al., 2016, study, where it was reported that the DE/A mutant reduced Claspin-Flag IP of Cdc7 ~5-fold, it was difficult to determine how quantifiably reproducible this effect was. The link between Cdc7 function and Chk1 activation in vivo is compelling, but the data concerning mechanism, in particular directness of Claspin as a Cdc7 target, were more difficult to glean.

Based on these comments and the points listed below, a revised manuscript is needed that addresses the biochemistry and as pointed out in point 3 below, tests additional cell lines to determine if the difference between cancer cells and normal cells with respect to Cdc7 and CK1γ1 kinases is robust.

Major points:

1) Figure 2. An important supporting line of evidence is that the AP_DE/A_ mutant (the acidic patch mutant that blocks Cdc7-phosphorylation of the Claspin CKBD) does not alter any other important functions of Claspin. For example, is this mutant still fully capable of CK1γ1-dependent activation of the checkpoint? What is the evidence that this mutation does not disturb other critical functions of Claspin needed to mediate the ATR activation of Chk1?

2) Is the S27A allele of Claspin defective in Chk1 phosphorylation after treatment of cells with HU, and does S27A block Cdc7 (and CK1) phosphorylation of Claspin? The manuscript (Figure 2) only presents Chk1 association with Claspin.

3) The authors' argument that Cdc7 is predominantly important for CKBD phosphorylation in cancer cells, whereas CK1γ1 is more important in non-cancer cells, is based on a single non-cancer line (NHDF). So it is essential to compare the relative importance of Cdc7 and CK1 for Chk1 phosphorylation in other such lines (e.g. RPE-1).

Other comments or clarifications:

1) Figure 1A. Claspin levels are lower in the siRNA of Cdc7 treatment. Why? This appears to be in contrast to Figures 3A and 3B.

2) The time course in Figure 1C should be extended since the difference at 10 hours is not striking.

3) Figure 1A. It is not clear what this siRNA really adds to this figure, when the key result is provided in Figure 1D, where constitutive depletion of Cdc7, via the promoter mutant, is shown to allow for normal S-phase progression of HCT116 cells yet insufficient activation of Chk1 upon replication stress. Wouldn't this Figure 1 be clearer with Figure 1D in place of Figure 1A? Also, how do you interpret why so much less Cdc7 is required for normal S-phase versus for responding to replication stress? Is it that Cdc7 is only one of multiple (e.g. CK1γ1) mechanisms to activate stress, so that stress activation depends potentially on multiple redundant pathways while origin function per se does not?

4) Related to point 1 above, are Claspin levels lower in the 323 cell line?

5) Figure 2C. The levels of expression of WT and mutant Claspin appear to be different and this may explain the difference in Chk1 activation.

6) The authors state "If the dephosphorylated oligopeptide was incubated with Cdc7 prior to pull down, the level of Chk1 pull down slightly increased (Figure 2—figure supplement 1C, compare lanes 8 and 9)." This claim is very hard to assess – the figure isn't clear, and the reader would need to see that the result is reproducible and quantified.

7) Does the DE/A mutant of Claspin also affect interaction of Claspin with CK1, or is this motif specific to Cdc7?

8) The text requires further work to improve clarity (e.g., a section entitled "Casein kinase also contributes to the phosphorylation of CKBD" is followed by a section called "Cdc7 is responsible for phosphorylation of CKBD". There are a number of minor textual errors, and abbreviations that are not clear (e.g. 'mAG-Claspin' in Figure 2C and associated text).

9) Some figure panels have very small text and are hard to read (e.g. bottom of Figure 1B, labels of flow cytometry data in Figure 1C, labels of flow cytometry data in Figure 4—figure supplement 2, sequences in Figure 5—figure supplement 1).

10) Is DNA replication as monitored by flow cytometry/BrdU still normal in the 323 cells (Cdc7 promotor mutant) upon treatment with siRNA to CK1? This is important to know, in order to interpret the effect on Chk1 phosphorylation.

11) Subsection “Cdc7 is responsible for phosphorylation of CKBD”: "Figure 5 and Figure 6—figure supplement 1B" should be "Figure 5 and Figure 5—figure supplement 1B".

12) The quantification of Claspin phosphorylation sites in Figure 5 and Figure 5—figure supplement 1 is very hard to assess. From what can be seen, it appears that the experiment was just done once, and so the significance of the findings remains unclear.

13) In Figure 6A and Figure 6—figure supplement 1, the authors argue that Cdc7 and CK1 have additive effects on phosphorylation of CKBD peptides, but the effects are not clear.

For example, a comparison of lanes 7-9 and 13-15 in Figure 6A does not support the authors claims that "Addition of Cdc7 in the presence of a low amount of CK1γ1 increased the phosphorylation level of the polypeptide. At a low level of Cdc7, the effect of CK1γ1 was additive on the level of the phosphorylation, whereas, at the highest concentration of Cdc7, similar levels of phosphorylation were observed regardless the presence or absence of CK1γ1 (Figure 6—figure supplement 1)."

14) The Discussion is essentially a repeat of the Results section with citation of figures and data all over again. It could be shortened and include discussion of previous data and the new observations.

---

## [Author Response]

[…]The evidence that Cdc7 directly interacts with the AP of Claspin and directly phosphorylates the Claspin CKBD is the weakest link in the study as it is difficult to glean how data were quantified. Even in the Yang et al., 2016, study, where it was reported that the DE/A mutant reduced Claspin-Flag IP of Cdc7 ~5-fold, it was difficult to determine how quantifiably reproducible this effect was. The link between Cdc7 function and Chk1 activation in vivo is compelling, but the data concerning mechanism, in particular directness of Claspin as a Cdc7 target, were more difficult to glean.

Several lines of evidence strongly indicate that Claspin is a direct target of Cdc7 kinase.

1) In vitro, whereas the purified wild-type Claspin binds to the purified Cdc7-ASK complex, APD_E/A_ mutant Claspin does not bind to it under the same condition (Yang et al. NC, 2016).

2) Whereas Cdc7 efficiently phosphorylates the purified wild-type Claspin, it does not phosphorylate the APD_E/A_ mutant Claspin to which it cannot bind.

3) The fifty amino acid polypeptide containing two CKBD motifs was phosphorylated by Cdc7 at the key serine/ threonine residues in vitro, albeit at a low level. The efficiency is low due to the lack of AP that recruits Cdc7.

We believe our Nature Communications paper (2016) clearly showed the role of AP in recruitment of Cdc7 in that AP_DE/A_ mutant of Claspin does not bind to Cdc7 both in vivo (293T expression) and in vitro (purified proteins) (Figure 2C and 3C).

Based on these comments and the points listed below, a revised manuscript is needed that addresses the biochemistry and as pointed out in point 3 below, tests additional cell lines to determine if the difference between cancer cells and normal cells with respect to Cdc7 and CK1γ1 kinases is robust.Major points:1) Figure 2. An important supporting line of evidence is that the APDE/A mutant (the acidic patch mutant that blocks Cdc7-phosphorylation of the Claspin CKBD) does not alter any other important functions of Claspin. For example, is this mutant still fully capable of CK1γ1-dependent activation of the checkpoint? What is the evidence that this mutation does not disturb other critical functions of Claspin needed to mediate the ATR activation of Chk1?

We reported followings in our previous paper (Yang et al. NC, 2016). AP_DE/A_ mutant is reduced in DNA replication and growth in normal cells, due to its inability to recruit Cdc7. However, AP_DE/A_ can bind to PCNA and Tim, which bind to Claspin through its N-terminal segments. It can bind to ATR as well. It binds to DNA with efficiency four times higher than that of the wild-type. Thus, it retains some activities of the wild-type Claspin. AP_DE/A_ is almost completely inactive in checkpoint activation in both MEF cells and U2OS cells. This is due to the loss of binding of both Cdc7 and CK1g1 in the AP_DE/A_ mutant.

In MEF, DNA synthesis (BrdU incorporation) is reduced by Claspin KO or by AP_DE/A_ mutation, probably due to reduced Mcm phosphorylation (NC paper, Fig. 3e). However, Claspin is not needed for Mcm phosphorylation in cancer cells, and thus AP_DE/A_ mutant does not show any S phase defect in for examples U2OS cells. In a paper from Steve Elledge’s lab, reduced cell numbers were reported in Claspin-depleted U2OS cells, but no significant effect on BrdU incorporation was seen. On the other hand, McGowan showed that replication fork rate is reduced but origin firing is more frequent in Claspin-depleted HeLa cells. Thus, loss of Claspin does not significantly affect the S phase progression per se. However, AP_DE/A_ exhibits defect in checkpoint activation in U2OS cells (Fig. 2C). Therefore, we believe that the effect of AP_DE/A_ mutation on checkpoint activation is due to its inability to recruit Cdc7, not through indirect effect of reduced DNA synthesis.

2) Is the S27A allele of Claspin defective in Chk1 phosphorylation after treatment of cells with HU, and does S27A block Cdc7 (and CK1) phosphorylation of Claspin? The manuscript (Figure 2) only presents Chk1 association with Claspin.

The ST27A mutation includes the three serine/ threonine residues (T916,S945,S982) in CKBD, and therefore, this mutant is expected to be defective in CHK1 phosphorylation after treatment of cells with HU. In fact, similar mutants were reported to be defective in checkpoint activation (JBC 281,33276[2006]; 3A[T916A,S945A,S982A] mutant). We have reported that the ST27A mutation is partially defective in Cdc7-mediated phosphorylation, suggesting that the mutated serine/ threonines include some but not all the Cdc7-directed phosphorylation residues (Figure 7a of Yang et al. NC, 2016). On the other hand, the same mutant was phosphorylated by CK1g1 as efficiently as the wild-type was (new Figure 6-figure supplement 2), suggesting that CK1g1 can phosphorylate the residues other than ST27.

3) The authors' argument that Cdc7 is predominantly important for CKBD phosphorylation in cancer cells, whereas CK1γ1 is more important in non-cancer cells, is based on a single non-cancer line (NHDF). So it is essential to compare the relative importance of Cdc7 and CK1 for Chk1 phosphorylation in other such lines (e.g. RPE-1).

In accordance with this comment, we have examined two more non-cancer cell lines, RPE-1 and TIG-3, and the new data show that checkpoint activation is dependent much more on CK1g1 than on Cdc7 in both cell lines (new Figure 5-figure supplement 4), supporting our argument above.

Other comments or clarifications:1) Figure 1A. Claspin levels are lower in the siRNA of Cdc7 treatment. Why? This appears to be in contrast to Figures 3A and 3B.

We have done siRNA depletion of Cdc7 many times, and generally did not see much effect on the Claspin protein level, more concomitant with the results of Figures 3A and 3B. One of such data are shown in new Figure1-figure supplement 1. The old data (the original Figure 1A) might have been the result of some experimental variations.

2) The time course in Figure 1C should be extended since the difference at 10 hours is not striking.

We have redone this experiment, took longer time course until 22 hrs after release. Similar to the previous data, we did not see difference in S phase progression between the parent HCT116 and Cdc7-promoter mutant HCT116-323 cells.

3) Figure 1A. It is not clear what this siRNA really adds to this figure, when the key result is provided in Figure 1D, where constitutive depletion of Cdc7, via the promoter mutant, is shown to allow for normal S-phase progression of HCT116 cells yet insufficient activation of Chk1 upon replication stress. Wouldn't this Figure 1 be clearer with Figure 1D in place of Figure 1A? Also, how do you interpret why so much less Cdc7 is required for normal S-phase versus for responding to replication stress? Is it that Cdc7 is only one of multiple (e.g. CK1γ1) mechanisms to activate stress, so that stress activation depends potentially on multiple redundant pathways while origin function per se does not?

We agree with the reviewer and took out the Figure 1A, and put this figure (after replacement with a new data) in Figure1-figure supplement 1. The reason why checkpoint activation is so sensitive to the level of Cdc7 in HCT116 is not clear. Cdc7 kinase needs to be somehow activated and/or relocated upon replication stress to phosphorylate the CKBD of Claspin which is presumably at the stalled replication fork. In cancer cells, the number of Cdc7 was estimated to be more than 1 million per cell. Even after 10-fold reduction, the numbers are still higher than the estimated numbers of replication origins, which may be sufficient for normal S phase progression. For initiation of DNA replication, Cdc7 may need to be recruited to any one of the origins present on the chromosome within the same replication timing domain, and thus the rate of encounter with the target would be high even at a low concentration of Cdc7. On the other hand, certain concentrations of Cdc7 would be needed for rapid recruitment of Cdc7 to the particular stalled replication fork on the chromosome, which is much lower in number.

4) Related to point 1 above, are Claspin levels lower in the 323 cell line?

As shown in Fig. 3B, Claspin protein level is similar in the parent and 323 cells.

5) Figure 2C. The levels of expression of WT and mutant Claspin appear to be different and this may explain the difference in Chk1 activation.

We have repeated this experiment. The new data in Figure 2C shows that checkpoint activation is much lower in AP_DE/A_ mutant in spite of similar levels of ectopically expressed Claspin.

6) The authors state "If the dephosphorylated oligopeptide was incubated with Cdc7 prior to pull down, the level of Chk1 pull down slightly increased (Figure 2—figure supplement 1C, compare lanes 8 and 9)." This claim is very hard to assess – the figure isn't clear, and the reader would need to see that the result is reproducible and quantified.

We agree with the comment and redone the experiment. The new data in Figure 2-figure supplement 1 show clear increase of Chk1 binding to the peptide upon phosphorylation by Cdc7. In the new experiment, we used 10 times more Cdc7 kinase, since the efficiency of phosphorylation is low on the peptide due to the lack of AP. Even under this condition, the binding is still inefficient due to incomplete phosphorylation of the peptide.

7) Does the DE/A mutant of Claspin also affect interaction of Claspin with CK1, or is this motif specific to Cdc7?

We have examined the interaction of AP_DE/A_ mutant with CK1γ1. As shown in Author response image 1, Claspin interacts with CK1γ1 as well, and this interaction is significantly reduced in the AP_DE/A_ mutant. This may also explain the almost complete loss of checkpoint activation in MEF cells with AP_DE/A_ mutant Claspin.

**Author response image 1. respfig1:** Flag-tagged wild-type (lanes 1 and 4) or APDE/A (lanes 3 and 6) or HA-tagged wild-type (lanes 2 and 5; negative control) Claspin was transiently expressed in 293T cells, and extracts were made with CSK buffer containing 0.1% Triton X-100. Immnoprecipitates with anti-Flag antibody were analyzed for CK1γ1 (lanes 4-6). Inputs were also analyzed (lanes 1-3).

8) The text requires further work to improve clarity (e.g., a section entitled "Casein kinase also contributes to the phosphorylation of CKBD" is followed by a section called "Cdc7 is responsible for phosphorylation of CKBD". There are a number of minor textual errors, and abbreviations that are not clear (e.g. 'mAG-Claspin' in Figure 2C and associated text).

In accordance with the comments, we have changed the text. We changed the order of the two sections pointed out by the reviewer above.

9) Some figure panels have very small text and are hard to read (e.g. bottom of Figure 1B, labels of flow cytometry data in Figure 1C, labels of flow cytometry data in Figure 4—figure supplement 2, sequences in Figure 5—figure supplement 1).

We also changed the sizes of the text in the figures.

10) Is DNA replication as monitored by flow cytometry/BrdU still normal in the 323 cells (Cdc7 promotor mutant) upon treatment with siRNA to CK1? This is important to know, in order to interpret the effect on Chk1 phosphorylation.

We show the effect of CK1g1 depletion in HCT116-323 cells and presented the new data in Figure 5-figure supplement 2. The average BrdU intensity of S phase cells is not affected by CK1g1 depletion in HCT116-323 cells, although the fraction of S phase cells is slightly reduced, suggesting that cells may be arrested at other phases of the cell cycle. The data indicate that DNA replication of S phase cells is not affected by CK1g1 depletion.

11) Subsection “Cdc7 is responsible for phosphorylation of CKBD”: "Figure 5 and Figure 6—figure supplement 1B" should be "Figure 5 and Figure 5—figure supplement 1B".

This was corrected.

12) The quantification of Claspin phosphorylation sites in Figure 5 and Figure 5—figure supplement 1 is very hard to assess. From what can be seen, it appears that the experiment was just done once, and so the significance of the findings remains unclear.

Experiments were conducted three times for HU, and HU+siCdc7 cells and twice for no treat and siCdc7 cells. We now present all the data in Figure 4 and Figure 4-figure supplement 1.

13) In Figure 6A and Figure 6—figure supplement 1, the authors argue that CDC7 and CK1 have additive effects on phosphorylation of CKBD peptides, but the effects are not clear.For example, a comparison of lanes 7-9 and 13-15 in Figure 6A does not support the authors claims that "Addition of Cdc7 in the presence of a low amount of CK1γ1 increased the phosphorylation level of the polypeptide. At a low level of Cdc7, the effect of CK1γ1 was additive on the level of the phosphorylation, whereas, at the highest concentration of Cdc7, similar levels of phosphorylation were observed regardless the presence or absence of CK1γ1 (Figure 6—figure supplement 1)."

We apologize for the confusion. The experiments of Figure 6 were conducted with a 50 aa polypeptide that is not phosphorylated by CK1g1 to any significant extent. Thus, as pointed out by the reviewer, we cannot argue for the additive effect from this data. Our argument is based on the results in Figure 6-figure supplement 1 in which we have conducted kinase assays with a longer Claspin-derived polypeptide #27 containing both CKBD and AP. This polypeptide can be phosphorylated by both Cdc7 and CK1g1.

The results indicate that the presence of the two kinases show additive effect on the phosphorylation of the polypeptide #27.

14) The Discussion is essentially a repeat of the Results section with citation of figures and data all over again. It could be shortened and include discussion of previous data and the new observations.

In accordance with this comment, we rewrote the Discussion, removing the repeat of the results and adding more discussion in relation to the previous data.